# Membrane marker selection for segmenting single cell spatial proteomics data

Monica T. Dayao [1,2], Maigan Brusko[3], Clive Wasserfall[3] & Ziv Bar-Joseph [2,4✉]

The ability to profile spatial proteomics at the single cell level enables the study of cell types, their spatial distribution, and interactions in several tissues and conditions. Current methods for cell segmentation in such studies rely on known membrane or cell boundary markers. However, for many tissues, an optimal set of markers is not known, and even within a tissue, different cell types may express different markers. Here we present RAMCES, a method that uses a convolutional neural network to learn the optimal markers for a new sample and outputs a weighted combination of the selected markers for segmentation. Testing RAMCES on several existing datasets indicates that it correctly identifies cell boundary markers, improving on methods that rely on a single marker or those that extend nuclei segmentations. Application to new spatial proteomics data demonstrates its usefulness for accurately assigning cell types based on the proteins expressed in segmented cells.

[1] Joint Carnegie Mellon University-University of Pittsburgh Ph.D. Program in Computational Biology, Pittsburgh, PA, USA. [2] Computational Biology Department, School of Computer Science, Carnegie Mellon University, Pittsburgh, PA, USA. [3] Department of Pathology, Immunology and Laboratory Medicine, University of Florida, Gainesville, FL, USA. [4] Machine Learning Department, School of Computer Science, Carnegie Mellon University, Pittsburgh, PA, USA. ✉email: zivbj@cs.cmu.edu

Recent advances in spatial proteomics enable the study of protein levels at the single cell resolution. Methods, including CODEX[1] and digital spatial profiling (DSP)[2], can detect the location of up to 70 different proteins in a single tissue section. Information about the levels of proteins in single cells can be used to answer questions regarding the set of cell types in a sample[3,4], the spatial distribution of these types[5–7], and the interactions between different cells and cell types in a tissue[8,9].

One of the first questions we need to address when analyzing spatial proteomics data is the identification of the location of cells in the sample, often termed cell segmentation. This differs from nuclei segmentation, which identifies just the nuclei of the cells. Cell segmentation is a crucial step since errors in identifying cells and their boundaries have a direct impact on our ability to correctly quantify the expression levels of proteins in these cells. Several methods have been developed for segmenting cells, and these often use one of two approaches. The first is to extend the segmentations of the nuclei, which are relatively easy to stain. Toolkits that use this approach include Cytokit[10], CellProfiler[11], and BlobFinder[12]. The second is to use a dedicated channel (often comprising a membrane protein) for cell segmentation[11,13,14]. These types of methods often use a Voronoi-based (centered at the nucleus)[15], graph cut[14], watershed transform[16], or deep learning[17] approach to identify boundaries between neighboring cells. While both of these approaches can lead to good results, they both suffer from potential downsides. The nucleus extension methods often assume that cells are densely packed. However, in cases where cells are not actually in physical contact, or when their shapes are irregular, such an approach can lead to incorrect segmentation. The membrane channel approach works well if the protein selected as marker is indeed expressed in all cells profiled. However, in many cases, it is not clear if a single protein can serve as a marker for specific tissue data, and even if a protein is a marker for some cells, it may not be expressed in others[6,18], leading to poor segmentation results for these cells.

A unique advantage of spatial proteomics is that, unlike prior methods, they do not require such markers to be selected in advance. Since these studies profile tens of proteins, a marker can be selected post-experiment. Moreover, there is no need to select a single marker. Even if a single marker does not work for all cells, a combination of such markers may be useful for segmenting all

cells. However, it is often challenging to know a-priori which combination would work best for the sample being profiled. An unsupervised correlation-based approach, suggested in[19], is one of the first attempts to use the image data to determine such markers. However, as we show, while that approach may work well in some cases, it does not work well with the datasets used here.

To select an optimal set of markers, we developed a method for RAnking Markers for CEll Segmentation (RAMCES). Following pre-processing, we use a pre-trained convolutional neural network (CNN) to identify the top markers for an unseen dataset. Next, we construct a new weighted channel using the top CNN predictions and use it for segmenting cells in the dataset. We tested RAMCES on CODEX datasets from three different organs which were profiled as part of HuBMAP[20] as well on the mouse spleen dataset from[1] and a cancerous bone marrow dataset from[7]. We compared RAMCES with sequence-based and correlation-based methods for determining membrane/surface proteins[19,21–23] and showed that RAMCES outperforms these methods. As we show, RAMCES correctly identifies relevant markers for the different tissues and is able to successfully use the integrated channel for accurate segmentation of cells.

## Results

We developed a method for RAnking Markers for CEll Segmentation (RAMCES) in CODEX data. RAMCES uses a CNN to identify a weighted combination of membrane marker proteins for new spatial proteomics datasets, regardless of the tissue or organ being studied (Fig. 1). RAMCES first learns to predict cell membrane markers by training a CNN on manually annotated CODEX data, where proteins are annotated as membrane markers or not (Figs. 1, 2). Given a CODEX dataset profiling a new tissue or region, RAMCES uses the learned CNN to rank proteins based on how likely they are to be membrane markers in the dataset. Next, it uses the ranking to generate new, combined images based on the top-ranked proteins. Finally, using RAMCES output, we can perform segmentation for the new data using a membrane-based segmentation method, such as Cytokit[10]. See "Methods" for details.

**RAMCES identifies combinations of membrane markers**. To test RAMCES, we first manually labeled four different CODEX datasets and used them for cross-validation testing (Supplementary

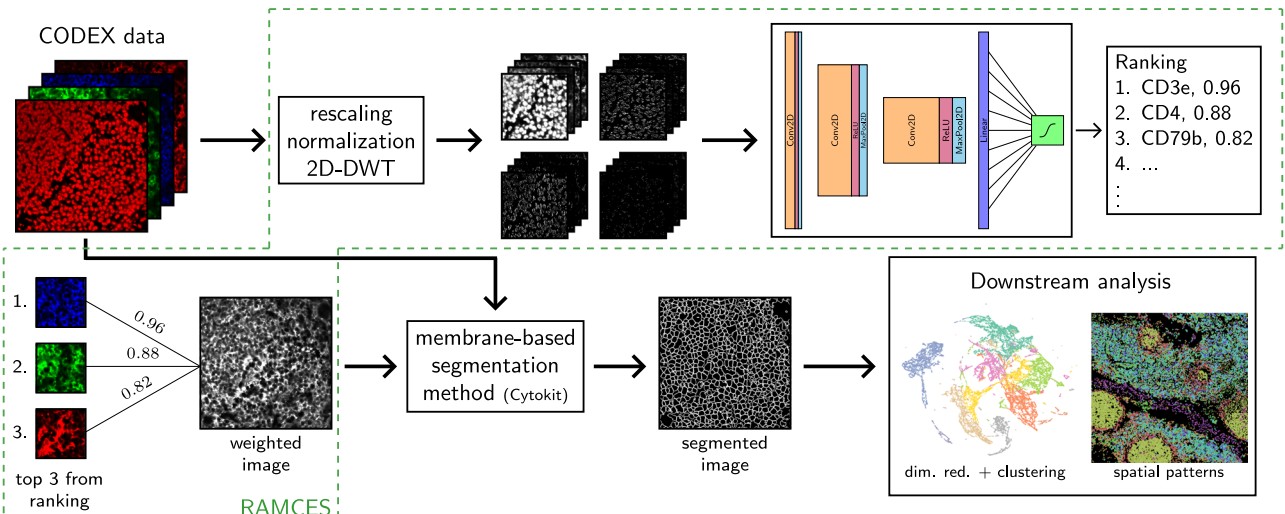

**Fig. 1 Overview of the RAMCES segmentation pipeline.** First, CODEX data goes through a series of pre-processing steps, including rescaling, normalization, and the discrete wavelet transform (DWT). Next, a pre-trained CNN is applied to the data to rank proteins/markers. Top markers are combined to create weighted images, which are then used for segmenting cells by a membrane-based segmentation pipeline, such as Cytokit[10]. The resulting segmented data can then be used for downstream analyses and for cell type assignment.

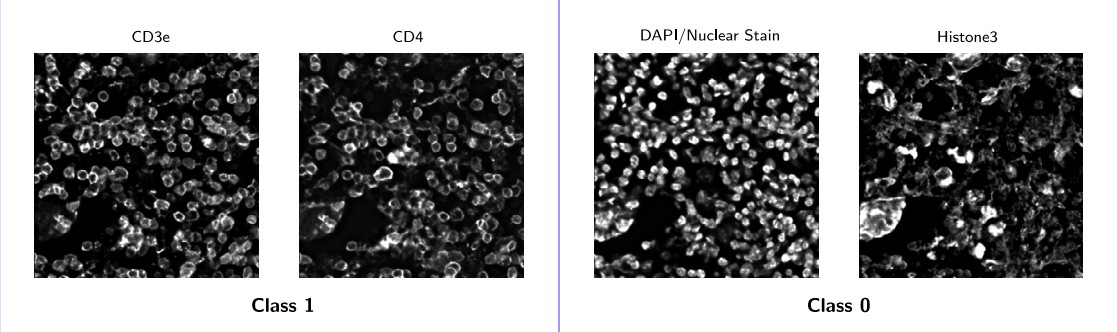

**Fig. 2 Examples of labeled CODEX proteins.** Proteins that qualitatively labeled membranes well were labeled as class 1, and proteins that labeled other cellular components were labeled as class 0.

Table 3). Our labeling selects a subset of the profiled proteins for each tissue and marks them as membrane markers (Fig. 2). Three of the datasets (2–4, Supplementary Table 1) were from thymus, lymph node, and spleen from the University of Florida. The fourth dataset (dataset 1) was from the mouse spleen[1]. These datasets span across different tissues, two different labs, and two organisms, which allows us to test both the generalization of our method to multiple regions and tissues and the use of our method to overcome lab-specific batch effects. We manually curated the proteins in each of the datasets and used these for training and testing ("Methods", Supplementary Table 1). Figure 3a presents the resulting receiver operating characteristic (ROC) and precision recall (PR) curves from 4-fold cross-validation. On average, our model achieves a ROC-AUC of 0.79. We additionally performed bootstrap analysis by training and testing the CNN model on 100 different bootstrap samples from datasets 1–4 ("Methods"). Results, presented in Supplementary Fig. 1, show that performance on bootstrap subsets is consistent, highlighting the robustness of the method.

RAMCES selects membrane markers based on the CODEX data itself. An alternative is to use a known list of membrane markers[21–23]. To compare to such an approach, we ranked markers using three popular sequence-based methods developed for predicting membrane or surface/signaling proteins. These include SURFY[21], which uses a random forest model to predict human surface proteins from sequence and domain-specific features, PrediSi[22], which uses a position weight matrix approach to predict signal peptide (SP) sequences, and SignalP 5.0[23], which uses a neural network-based approach to predict signaling peptides from amino acid sequences. Each of these methods assigns a score between 0 and 1 to every protein, with a higher score indicating that the protein is a surface marker or a signaling protein. We also compared RAMCES to another unsupervised method that is based on the input image data. This method uses Spearman's correlation to rank protein pairs and selects proteins appearing in the top-ranked pairs[19] ("Methods").

Figure 3b presents the resulting ROC and PR curves for the classification performance of all methods on datasets 2–4. The dataset from[1] was left out in this comparison because the SURFY method[21] considers only human proteins. As can be seen, our context-specific CNN method outperformed all of the other methods. Specifically, our method achieved an ROC-AUC of 0.81 which is 20% higher than the best sequence-based or correlation-based method for this data (SignalP5, 0.68).

Table 1 lists the top 5 proteins identified by the CNN for each held-out dataset from cross-validation. Bolded protein names are either membrane or cell surface proteins, as defined by the Human Protein Atlas[24] or the Cell Surface Protein Atlas (CSPA)[25]. All but Ki67, which was ranked 5th by our model in the lymph node dataset with a score of 0.544, are defined as membrane or cell surface proteins, indicating that the CNN can

accurately identify markers for cell segmentation. This result is especially promising when considering the murine spleen dataset from[1]; this dataset is from a different lab and organism than the other datasets used for cross-validation.

We additionally computed protein rankings on datasets 5–11 using the cross-validation model trained on datasets 2–4. These rankings are shown in Supplementary Table 4. The top 3 proteins for these datasets are labeled as membrane proteins by the Human Protein Atlas[24], with the exception of spleen dataset 9. In dataset 9, only CD20 was given a score greater than 0.5 (0.787). All other proteins in this dataset received scores less than 0.5, indicating that the model classified these proteins as non-membrane labeling proteins. Supplementary Figure 2 shows an example tile for each of the top 5 proteins in dataset 9, which shows that only CD20 looks like it labels cell membranes well.

**Pre-processing with DWT improves CNN performance**. We also evaluated the pre-processing steps performed by RAMCES. For this, we trained models with and without using the discrete wavelet transform (DWT) on the input images for 500 epochs. For this evaluation, we used the murine spleen dataset[1], which we split 70/30 into training and test sets. Supplementary Figure 3 shows the ROC and PRC curves resulting from evaluating the two models on the test set. Using the DWT improves the performance of the classification model from an ROC-AUC of 0.874 to 0.924 and a PRC-AUC of 0.789 to 0.909.

**Comparing segmentation methods**. We next used a weighted combination of the top-ranked proteins by RAMCES to segment CODEX data. Here we report the results for datasets 5–7 (Supplementary Table 1). For this, we created new images which use weighted expression levels for the top 3 proteins for each dataset, where the weights are proportional to the CNN confidence in labeling them as membrane markers ("Methods"). Selected proteins are listed in Supplementary Table 4, which also lists the output RAMCES scores and entropy values to give a measure of the uncertainty of the model. The new image was subsequently segmented using Cytokit to obtain cell boundaries for each of the datasets[10,15] ("Methods"). Next, we compared several segmentation methods to evaluate the resulting segmentation of our method, which we refer to as the 'Top 3' segmentation. Specifically, we compared the resulting segmentation to (1) the Cytokit segmentation that uses only the top-ranked protein marker for segmentation (Top 1) and (2) Cytokit's default segmentation which is based on extending the nucleus segmentation by a specified radius (Nucl-ext). Example segmented cells, overlaid with the weighted membrane image from RAMCES, are shown in Fig. 4. The Top 3 segmentation, in green, follows the outlines of the cell membranes more closely than the other methods. In areas

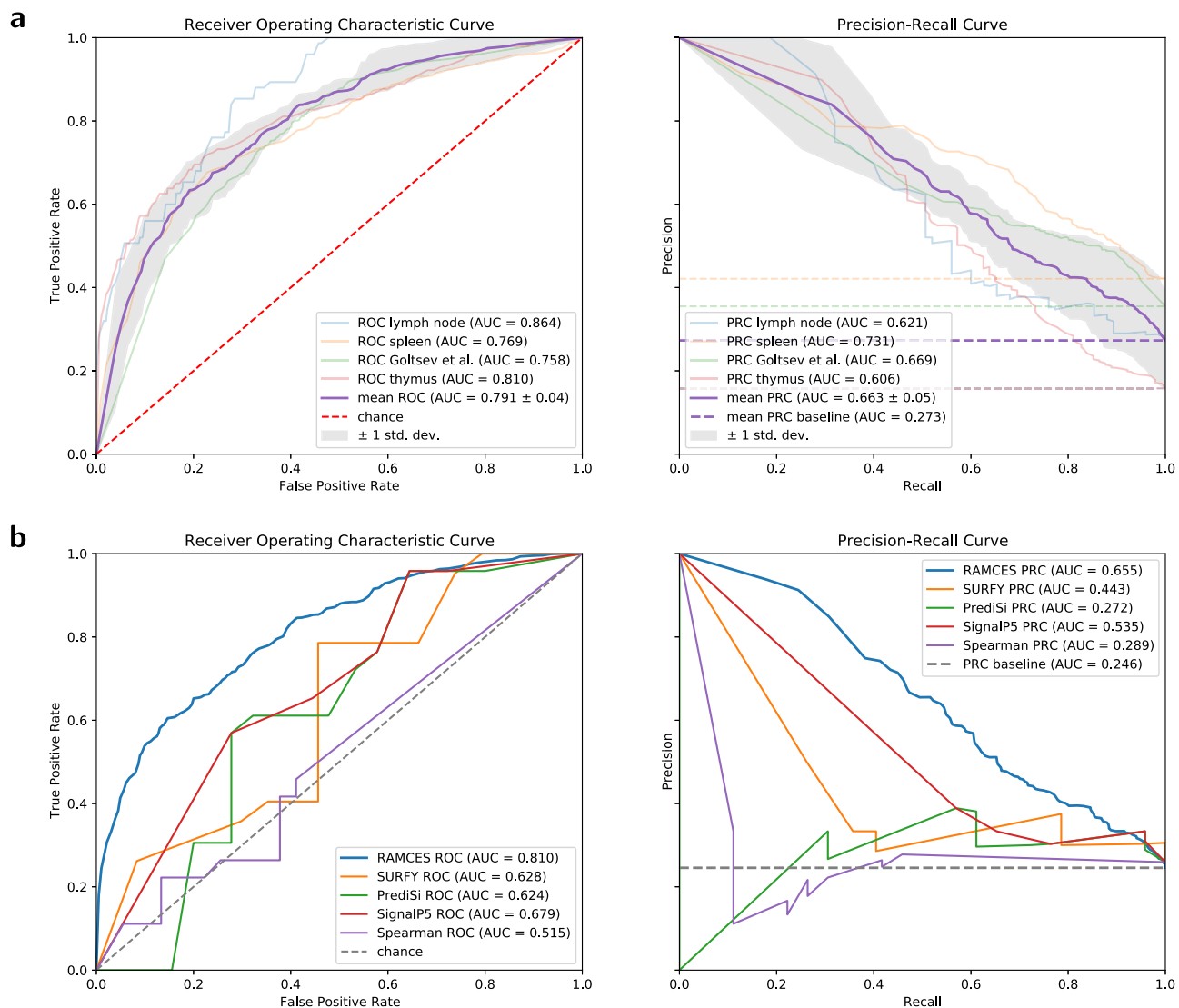

**Fig. 3 Cross validation and comparison receiver operating characteristic (ROC) and precision-recall (PR) curves evaluating the results on the binary classification of membrane protein markers. a** ROC and PR curves for the RAMCES CNN model from cross validation runs. Dashed lines represent the random chance baselines for the ROC and PR curves. The mean ROC and PR curves (purple) are presented with error bands representing ± standard deviation ($n = 4$ cross-validation models). **b** Average ROC and PR curves comparing RAMCES with other membrane protein prediction methods, based on results from datasets 2–4 (Supplementary Table 1).

**Table 1 Top 5 ranked proteins from each cross validation model.**

| Rank | Lymph node (2) | Spleen (3) | Thymus (4) | Murine spleen (1) |
|---|---|---|---|---|
| 1 | **CD3e** | **CD45** | **CD4** | **CD79b** |
| 2 | **CD8** | **CD4** | **CD3e** | **IgD** |
| 3 | **CD45** | **CD3e** | **CD8** | **CD44** |
| 4 | **CD4** | **CD45R0** | **CD20** | **CD45** |
| 5 | Ki67 | **CD8** | **CD45R0** | **CD90** |

Column headers indicate tissue and dataset number from Supplementary Table 1. Bolded protein name means that is labeled as a membrane protein by the Human Protein Atlas[24] for datasets 2–4. For the murine dataset, bold means it is labeled as a membrane protein by the CSPA[25] with 'high confidence'.

where the weighted membrane image (in red) is empty, the segmentation follows the nuclei instead (in gray).

To quantitatively evaluate the difference between the segmentations, we selected the following protein sets to examine cells for coexpression according to the different segmentations: CD3+CD4+ for CD4+ T cells, CD3+CD8+ for CD8+ T cells, CD4+CD8+ for double-positive T cells, and CD68+CD4−CD8− for macrophages. '+' and '−' correspond to the presence and absence of a protein, respectively. These cell types are expected to be well represented in lymph node, spleen, and thymus[26,27]. We calculated the percentages of cells in each dataset that match the protein coexpression sets (Supplementary Table 5 and Supplementary Fig. 8, "Methods"). The percentage of cells that coexpress these sets of proteins increases when using RAMCES segmentation compared to the default nucleus extension segmentation from Cytokit, which shows that there is a quantitative difference between the two segmentation methods.

*Comparing RAMCES to individual markers and manual segmentation.* To further test the usefulness of combining multiple markers, we computed the agreement between RAMCES segmentation and segmentation based on each of the top three channels individually ("Methods"). Results, presented in Supplementary Table 6, show that RAMCES obtains, on average, a much better overlap with each of the three top individual

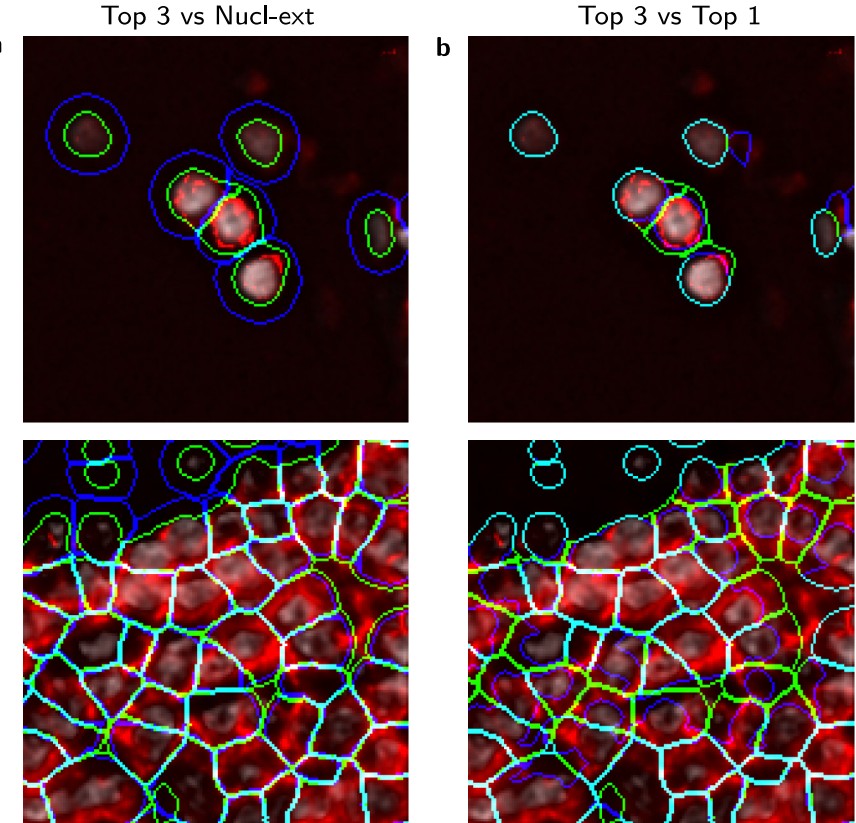

**Fig. 4 Segmentation comparisons. a** Comparison of segmentations (from dataset 5, Supplementary Table 1) between RAMCES with the top 3 proteins combined (Top 3) and the default Cytokit[10] nucleus extension method (Nucl-ext), and **b** between using the top single (Top 1) and top 3 ranked membrane markers. In each image, green contours are the Top 3 segmentation, and blue contours show the segmentation we are comparing to. Cyan indicates that the two segmentations overlap. The red channel shows the combined weighted membrane image from the top 3 ranked proteins, and the gray channel shows the DAPI (nucleus) stain. The top images come from tile X = 4, Y = 5, and the bottom images come from tile X = 4, Y = 4 of dataset 5.

channels when compared to their average pairwise overlap (improvement of 6.5% using Jaccard and 3.5% using Dice), highlighting the advantage of the weighted channel generated by RAMCES.

Since no segmentation ground truth is available for these data, we had two experts manually segment cells for two image tiles and compared RAMCES segmentation performance (using combined markers) to segmentation performance using only a single marker and to the nucleus extension method ("Methods"). For this, we used two image tiles from lymph node dataset 5. Manual segmentation was performed by referencing all of the image channels; however, manual segmentation is biased since it relies on prior assumptions[28], and in this case, the DAPI and CD45 channels were most heavily relied on by the experts we used. Supplementary Table 9 shows that the two expert segmentations have an average Dice coefficient of 0.6987, which is in line with prior work from[28]. Results from comparing different segmentation methods with the first expert annotation, presented in Supplementary Table 7, show that on average, RAMCES segmentation combining the top 3 (and top 2 and 4) ranked proteins improve on the average performance of the top three ranked markers (improvement of 2.1% using Jaccard and 1.3% for Dice). We observe that the 2nd ranked marker on its own performs slightly better than RAMCES; however, we note that this marker is CD45, which is the one used by the expert to segment the cells. RAMCES segmentation also outperforms the nucleus extension segmentation by 8.1% (Jaccard) and 5.1% (Dice) on average. Supplementary Table 8 shows a similar comparison to annotations from the second expert.

We additionally performed an analysis comparing the agreement and disagreement areas between the different segmentations. Specifically, we looked at pixels in the image that RAMCES assigns to the inside of cells and other methods assign to the background, and at pixels that RAMCES assigns to background and the other segmentation methods assign to the inside of cells (Supplementary Figs. 4–7). We would expect that successful methods would have an average biomarker distribution for areas where the method assigns to inside cells and the comparison method assigns to background similar to the distribution for areas where both methods assign to inside cells (Supplementary Fig. 4). Supplementary Figure 5 compares the RAMCES segmentations with the nucleus extension segmentations and highlights that there is increased biomarker signal in pixels identified as inside cells by both of the segmentation methods when compared to those determined to be outside cells, as we would expect. Supplementary Figure 6 shows that in areas which RAMCES labels as inside of cells and the nucleus extension method does not, the biomarker signal resembles that of levels seen for agreement cell (foreground) areas. In contrast, areas where the nucleus extension method labels as cells and RAMCES does not look much more similar to background areas. This result suggests that the RAMCES segmentations capture more of the biomarker signal than the nucleus extension method, which we can interpret as a measure of improved segmentation. Supplementary Figure 7 summarizes the disagreements between RAMCES and the first manual expert segmentations in a similar way to the comparison in Supplementary Fig. 6. This again shows that for all biomarkers except DAPI, the RAMCES

segmentation actually agrees better with cell biomarker signal than the manual segmentations.

**Application to new CODEX datasets**. We performed additional CODEX experiments to test the ability of RAMCES to generalize to new data. We first tested RAMCES on new data we generated profiling lymph node, spleen, and thymus using additional protein markers (29 distinct markers, datasets 8–10, Supplementary Table 1). We also analyzed CODEX data for another tissue, cancerous bone marrow from[7]. This dataset profiles 59 distinct markers (dataset 11, Supplementary Table 1). To segment these datasets, we used RAMCES with the CNN model trained on datasets 2–4, which profiled only 19 distinct markers. As before, we rank proteins for these datasets using the CNN model and select the top 1–3 as membrane markers for the segmentation pipeline ("Methods"). Selected proteins agree well with known membrane markers for these tissues (Supplementary Table 4).

Next, we used the protein levels assigned to each segmented cell to assign cell types to the data. For this, we quantified the average intensity levels for each protein in each cell using Cytokit, followed by Leiden clustering of all cells ("Methods"). In total there were 77576, 103010, 84207, and 5276 cells for the thymus (10), lymph node (8), spleen (9), and cancerous bone marrow (11) datasets, respectively. We next looked at top markers for each cluster and assigned cell types using these markers. Results for thymus dataset 10 are presented in Fig. 5. Results for datasets 8 and 9 are presented in Supplementary Fig. 9 and dataset 11 in Supplementary Fig. 10. The CODEX panel for these datasets 8–10 was designed to delineate cell types within clusters using canonical markers, namely, pan-cytokeratin (PAN-CK) for epithelial cells, smooth muscle actin (SMActin), CD31, and LYVE-1 for lymphatic and vessel endothelial cells, and CD45 for immune cells of hematopoietic lineage. Among CD45+ cells, we further defined major cell subsets as follows: CD20+ B cells, CD3+CD4+ and CD3+CD8+ T cells, CD4+FOXP3+ regulatory T cells, and CD11c+CD68+CD15+ myeloid cells. Clusters denoted as proliferating cells expressed high Ki67. Additional markers incorporated in future panels will allow the classification of effector populations of both innate and adaptive immune cells. Figure 5 displays the spatial assignments associated with the different cell types for the thymus dataset along with a couple examples of labeled regulatory T cells, which clearly show expression of FOXP3. For the bone marrow dataset from[7], major cell types were defined as follows: CD7+CD8+ T cells, PAX5+Ki67+CD34+ proliferating stem cells, CD31+ endothelial and stromal cells, CD45RO+CD25+CD3+Ki67+ memory activated T cells, CD68+CD163+ monocytes and macrophages, and CD3+FOXP3+CD25+CD4+ memory regulatory T cells. Supplementary Figures 11–15 show UMAP embeddings colored by select marker proteins and channel montages for example cell types for datasets 8–11.

## Discussion
Cell segmentation has been a major challenge in computational biology for a number of decades now. Several methods have been proposed for this task, and these either extend nucleus segmentation or use a pre-determined membrane or cell surface marker for the task[10–14]. However, when profiling tissues with several different types of cells from different individuals, it is usually not possible to select markers that would work well for all cells in the sample.

A unique advantage of single cell spatial proteomics, in which we profile both the location and the level of proteins at high resolution, is the ability to select such markers in a post hoc rather than an ad-hoc manner. Since these technologies often profile tens of proteins, it is likely that among them we would identify a combination that would be best for the tissue/sample we are profiling.

To automate the identification process, we developed a new computational pipeline, RAMCES, that relies on deep neural networks to select the most appropriate markers for each dataset. Once trained, the method can be applied to new datasets even if none of the images in these datasets were manually annotated. Thus, the pipeline fully automates the process of marker selection for cell segmentation, drastically reducing the manual labor required to annotate new membrane channels for each new dataset. The trained RAMCES model used for evaluation in this paper is available at github.com/mdayao/ramces. Additionally, if it is known a priori that particular markers should not be used for segmentation, they can be removed from the RAMCES ranking and not combined in the final combined RAMCES output.

We applied RAMCES to several CODEX datasets from multiple tissues. As we show, outputs from RAMCES are used to successfully segment the cells, improving on methods that rely on known markers and those that directly extend the nucleus segmentation. Analysis of the expression of known markers in cells indicates that RAMCES is able to identify more cells expressing a combination of markers expected for the different tissues. Comparisons to manual segmentation in two tiles of lymph node dataset 5 showed that by using a combined channel, RAMCES improves on the average agreement achieved by the the top three markers, and it also improves on methods that use only the nuclear channel and its extension. Segmentation using only the 2nd ranked channel (CD45 for this dataset) has slightly better agreement with the manual segmentation than any of the combined segmentations. This is likely because, as mentioned, the experts referenced the CD45 channel most often when manually segmenting the tiles. This highlights the challenge with obtaining ground truth since any manually-curated ground truth is likely biased by the individual performing the segmentation[28]. By combining the top three channels, RAMCES is able to overcome such biases and obtain results that are in good agreement with both expert curation and expected cell populations. Application of the method to new datasets profiling 50% more proteins than the training datasets indicates that the method can scale well for new, unseen, proteins while maintaining accurate cell segmentation results. Additionally, application to a cancerous bone marrow dataset demonstrated the method's ability to generalize to tissues different than those in its training set.

We used the segmented cells to cluster and assign cell types for a number of new CODEX datasets from three different tissues. Results for the thymus, presented in Fig. 5, indicate that with RAMCES, we were able to observe the expected cellular organization and architecture of the thymus. Specifically, fields of CD4 and CD8 double-positive developing T cells (teal) are distributed throughout the thymic cortex. These double-positive T cells then interact with thymic epithelial cells and antigen-presenting cells (blue) in the medulla, driving their selection as single positive T cells (red–orange). Thus, the segmentation accurately captured expected expression profiles and cellular distribution.

While RAMCES was successful in the segmentation of samples from a number of different tissues, including lymph node, spleen, thymus, and cancerous bone marrow, more analysis is required to make sure that it can generalize to other tissues as well. We note that since the resulting RAMCES segmentation in this paper is dependent on Cytokit's segmentation, which is deterministic, we cannot derive statistics on issues related to false discovery rates (e.g., for cell identification) for different thresholds. In addition, while we believe the approach can be easily generalized to other types of spatial proteomics data, the current analysis only on

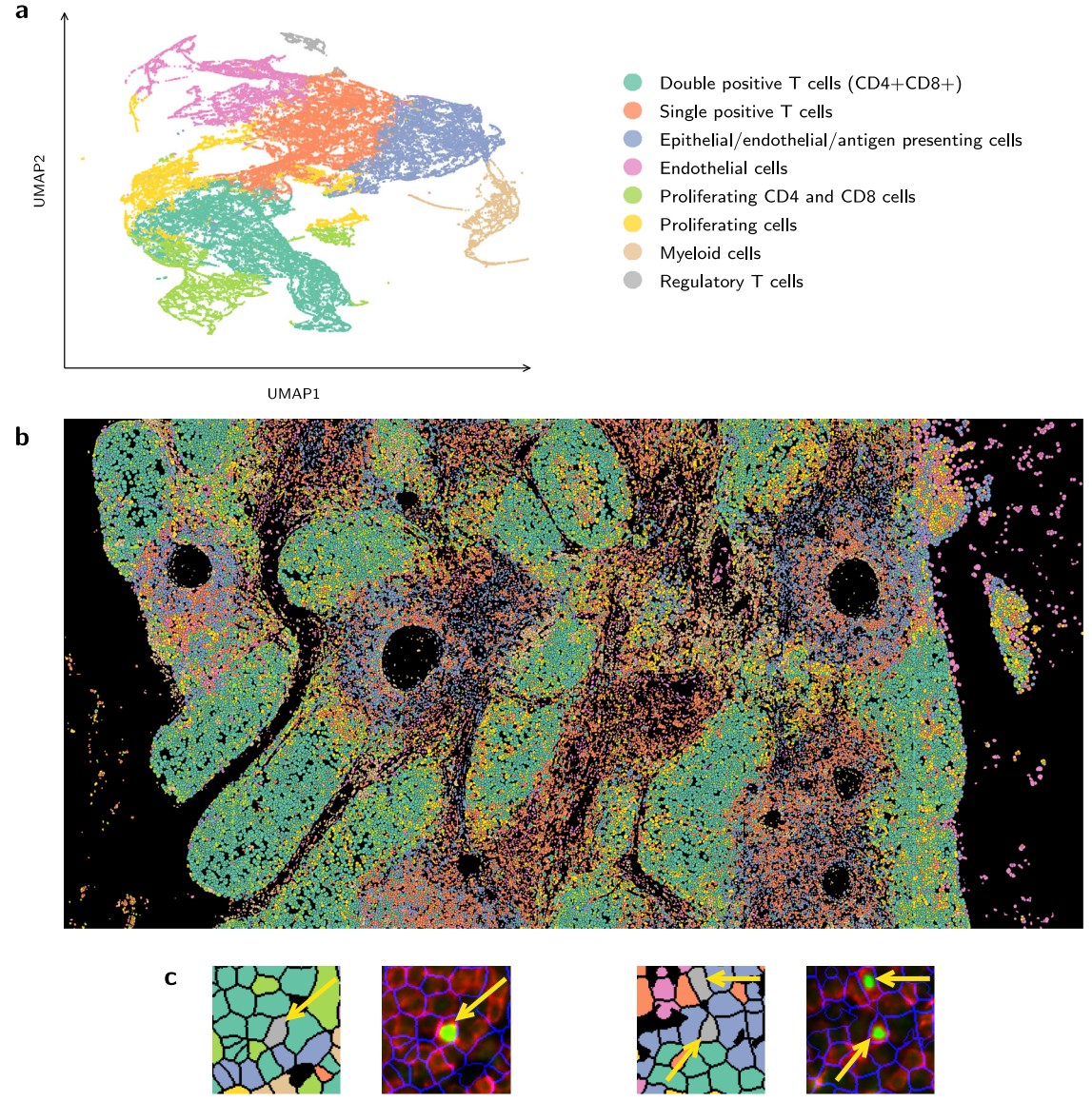

**Fig. 5 Spatial assignment of cell types. a** Clustering and UMAP visualization of cells based on the expression of profiled proteins, performed by the Cellar tool[41]. Cell types were assigned based on known markers ("Methods"). **b** Stitched tiles containing segmented cells from thymus dataset 10 (Supplementary Table 1) using the same colors as the cell type colors above. **c** Zoomed-in regions of (b) adjacent to their corresponding CODEX images, showing example cells labeled as regulatory T cells, colored gray and indicated by the yellow arrows. CODEX images show the cell segmentation outline (blue), CD4 (red), and FOXP3 (green).

focused CODEX which is the most widely used technique right now for HuBMAP[20].

Given the growing interest and use of spatial proteomics in several different studies, we believe that RAMCES can address an important need and will improve the ability to segment cells and assign accurate cell-based expression profiles.

## Methods

**Overview**. The initial method used by HuBMAP for segmenting CODEX data is based on Cytokit[10], an image cytometry toolkit for fluorescent microscopy datasets. Cytokit uses a pre-trained U-Net[29] deep neural network for nuclei segmentation. To extend the nuclei segmentation to full cell segmentation, Cytokit uses one of two options: (1) the nuclei acts as a center for each cell and is grown to a specified radius or until it collides with other cells, or (2) use (if known) a specified image channel for a membrane marker/protein in the dataset with a Voronoi-based method[15] to label the membranes of the cells. However, it is often unclear which protein best labels the cell membranes as this property differs between tissues and various regions within them and even between cells in the same tissue[30].

To address these problems, we developed a method for RAnking Markers for CEll Segmentation (RAMCES), which uses a CNN to identify a weighted

combination of membrane marker proteins for new CODEX datasets, regardless of the tissue or organ it profiles. The combined set of channels is then used to create a composite image which is fed into the segmentation pipeline (Fig. 1).

**Donor acquisition**. Organ donor tissue samples were recovered by the HuBMAP Lymphatic System Tissue Mapping Center (TMC) according to established protocols (https://doi.org/10.17504/protocols.io.bsdsna6e)[31] approved by the University of Florida Institutional Review Board (IRB201600029), the United Network for Organ Sharing (UNOS), in accordance with federal guidelines, and with written informed consent from each donor's legal representative. The studies were conducted in accordance with the relevant criteria set forth in the Declaration of Helsinki. Donor demographic information is available on the HUBMAP portal (https://portal.hubmapconsortium.org/) through the dataset IDs provided in Supplementary Table 1.

Human spleen, lymph nodes, and thymus were obtained from organ donors and processed within 16 h of cross clamp. Residual fat or connective tissues were removed from each tissue, and size and specimen recovery location and orientation noted and recorded in https://hubmapconsortium.github.io/ccf-ui/. 1 cm³ blocks were dissected from each tissue with known and registered location within the human body. Tissue cassettes were placed in at least 20 volumes of 4% PFA (i.e., 20 mL 4% PFA per 1 mL tissue) for 20–24 h, transferred to 70% ethanol, and

paraffin embedded in an automated tissue processor (Sakura VIP) within 3 days. Tissues were sectioned to 5 µm and stained with hematoxylin and eosin (H&E) and evaluated by an independent pathologist to assess organ normality.

**CODEX staining and imaging of human samples**. Barcoded antibody staining of tissue sections mounted on cover slips was performed using a commercially available CODEX Staining Kit according to the manufacturer's instructions for FFPE tissue (Akoya Biosciences) and as recorded for HuBMAP Lymphatic System TMC (https://doi.org/10.17504/protocols.io.be9pjh5n)[32]. Images were acquired at 20X magnification using a Keyence BZ-X810 microscope with a metal halide light source and filter cubes for DAPI (358), TRITC (550), CY5 (647), and CY7 (750). Typical images are 7×9 (3.77 × 3.58 mm) including acquisition of 17 Z-stack images at 1.5 µm pitch. Raw images were collected using the CODEX Processor software (version 1.30.0.12). Drift compensation, deconvolution, z-plane selection, and stitching were performed using the Cytokit software[10], using the Docker[33] container found at https://hub.docker.com/r/eczech/cytokit (version 'latest' uploaded on Feb 5, 2020). The display lookup table (LUT) for all figures presented in this paper is linear and covers the full range of data.

**Datasets**. A CODEX[1] dataset consists of 2D images of a tissue sample at a single-cell resolution, with each channel of the image visualizing the expression of a specific protein from fluorescent antibody probes. Other fluorescent stains, such as DAPI or DRAQ5 stains, can also be visualized. The images are obtained in tiles across the tissue sample at different z-levels; however, for our analysis, we use only the z-level with the best focus for each tile for downstream analysis. The best z-level is chosen by Cytokit[10] using a deep-learning-based classifier that scores image quality[34].

For the analysis presented in this paper, we collected data from several different tissues. For training and cross-validation of the neural network model, we used three human tissue datasets (lymph node, spleen, thymus) from the University of Florida and a previously published mouse spleen dataset from[1] (Supplementary Table 1). Supplementary Table 10 gives the table of antibodies used in the human tissue datasets. We used the trained model to choose membrane proteins from other CODEX datasets which profiled human tissues from thymus, lymph node, and spleen. We also used the model to choose membrane markers for a previously published dataset of cancerous bone marrow (dataset 11, Supplementary Table 1). This dataset was from region 4 of the 'Multi-tumor TMA' data from[7].

The training datasets and 3 of the testing datasets from the University of Florida (2–7, Supplementary Table 1) profiled 19 distinct markers. The other 3 datasets from UF (datasets 8–10) profiled 29. The dataset from[1] (dataset 1) profiled 31 distinct markers, and the bone marrow dataset from[7] (dataset 11) profiled 59. Supplementary Table 2 lists these proteins for each dataset. Each tile within a dataset spanned 1008 × 1344 pixels, except for dataset 11, which consisted of a single 1440 × 1920 tile. A more detailed specification of each of the datasets can be found in Supplementary Table 1. Throughout this paper, we refer to each dataset by its 'Dataset' number in Supplementary Table 1.

For training, each marker from training datasets 1–4 were manually labeled as one of two classes: labeling cell membranes and not labeling cell membranes (Fig. 2). Supplementary Table 3 lists these manual annotations.

**Convolutional neural network model to classify CODEX proteins**. We developed a CNN model to predict weighted combinations of membrane markers for a given CODEX dataset. To predict such proteins, the model uses the training data to learn a classifier for predicting: (1) for a cell membrane protein, or (0) for non cell membrane protein. The CNN includes three convolutional layers, each followed by a Leaky ReLu[35] activation function and a max-pooling operator, and a fully-connected layer with dropout[36] ($p = 0.3$) for better generalizability. The Leaky ReLU is a type of rectified linear activation function which is shown to increase model performance overall compared to sigmoidal or tanh activation functions[35]. It is defined as

$$f(x) = \begin{cases} 0.01x & x < 0 \\ x & \text{otherwise} \end{cases} \quad (1)$$

The last fully-connected layer is fed into a sigmoid activation function such that the output from the CNN is a value between 0 and 1, representing a score of how well the input image labels membranes. The sigmoid function is defined as

$$S(x) = \frac{1}{1 + e^{-x}} \quad (2)$$

*Pre-processing*. Before inputting the images into the CNN to predict membrane markers, RAMCES performs a series of pre-processing steps. First, each CODEX image tile is separated into its individual protein channels, such that each channel for each tile is a separate training sample. Only the unique protein channels are considered; blank channels and duplicates are removed. The samples are then rescaled to 1024 × 1024 pixels, z-normalized, and clipped at $\pm 3\sigma$, where $\sigma$ is the standard deviation of the pixel intensity values, to remove any extreme outliers. The 1024 × 1024 sized tiles are then split up into non-overlapping tiles of 128 × 128 pixels each, resulting in 64 separate images for each tile in the dataset. This yields 330,396 training images before data augmentation. For data augmentation during

training, the samples are randomly flipped (vertically and horizontally), rotated (by multiples of 90°), and translated.

A 2D DWT is then used on each sample. The DWT is commonly used for image compression and image denoising. It decomposes a signal into a set of mutually orthogonal wavelet basis functions by passing it through a series of filters. These filters can be denoted $g(n)$ and $h(n)$ for the low-pass and high-pass filters, respectively. In the 1D case, given a discrete signal $x(n)$ the output of the DWT can be written as the sub-sampled convolution of the signal and the two filters, yielding a low-pass and high-pass output that together is equivalent in size to the original signal:

$$y_{\text{low}}(n) = \sum_{k=-\infty}^{\infty} x(k)g(2n-k) = (x * g) \downarrow 2 \quad (3)$$

$$y_{\text{high}}(n) = \sum_{k=-\infty}^{\infty} x(k)h(2n-k) = (x * h) \downarrow 2 \quad (4)$$

For images, we perform the 2D DWT, which uses the 1D DWT on the rows and subsequently on the columns of the image array. This process yields four sub-band images: LL, LH, HL, and HH. The LL sub-band image is the downsampled version of the original image: the result of a low-pass filter in both the rows and the columns. For a multi-level DWT, the LL image would be used as input to the next level. The LH sub-band image is the result of a low-pass filter along the rows and a high-pass filter along the columns; this image contains the horizontal features of the original image. Similarly, the HL sub-band image isolates vertical features. The HH sub-band image is a result of the high-pass filter along both the rows and the columns, capturing most of the noise in the image. We show that using the 2D DWT improves the performance of our CNN (Supplementary Fig. 3), likely by isolating the edge-like features in the images. We used the Debauchies 2 wavelet from[37] to perform the 2D DWT on the CODEX images.

*Training*. The model was trained with CODEX datasets (1–4) as specified in Supplementary Tables 1 and 3 for 300 epochs. We used stochastic gradient descent[38] with a learning rate of 0.01 as the optimizer and binary cross entropy as the loss function. Each cross-validation model held out one of the four datasets for testing and trained on the other three. For the bootstrap analysis, we randomly sampled from datasets 1–4 with replacement to create a training set size equal to 60% of the total dataset size. The testing set comprised of the remaining samples. This process was repeated 100 times to obtain the 100 bootstrap models. We trained the model using the PyTorch deep learning framework[39] and visualized training metrics with Weights & Biases[40].

**Scoring and weighted images**. For the CODEX datasets used for evaluation (Supplementary Table 1), the score for each protein is calculated as follows. Each 1024 × 1024 pixel tile is split into 64 smaller subtiles, as we do for the training data, and a score between 0 and 1 is assigned for each of these smaller subtiles by the CNN. The highest score given to one of the 64 subtiles is used as the score for the entire tile. The reason we used the max is because several subtiles may contain few or no cells. In contrast, the majority of large tiles had several subtiles that did contain cells and so using the max leads to selecting one of the denser subtiles in the image. The final score for each protein is the average of the scores given to that protein across the larger tiles in the dataset.

To combine multiple proteins into a new image channel, we take the weighted average of the corresponding protein images (using their intensity values), weighted by the individual protein scores. More specifically, the $j$th pixel in the weighted image ($x_{\text{weighted}}^{(j)}$) is calculated from the intensity value of the $j$th pixels in the images of the selected proteins, with $w_i$ being their score.

$$x_{\text{weighted}}^{(j)} = \frac{\sum_{i=1}^{n} w_i x_i^{(j)}}{\sum_{i=1}^{n} w_i} \quad (5)$$

For the results shown here, the top $n = 3$ proteins were chosen to create the weighted image channel, as ranked by their score (1 is highest, 0 is lowest). See Table 1 and Supplementary Table 4 for the proteins used for each dataset and their scores. Supplementary Table 4 also lists the Shannon entropy for each protein, which can be interpreted as the uncertainty of the RAMCES CNN model. The Shannon entropy is calculated as

$$H = -p\log_2 p - (1 - p)\log_2(1 - p) \quad (6)$$

where $p$ is the output RAMCES score for that protein.

**Cell segmentation and post-processing**. Cell segmentation was performed using the Cytokit software[10], using the Docker[33] container found at https://hub.docker.com/r/eczech/cytokit (version 'latest' uploaded on Feb 5, 2020). The cell segmentation method by Cytokit[10] takes as input the CODEX image data, a nucleus channel, and a specified membrane channel (output from RAMCES in our case). If no membrane channel is specified, the default nucleus extension segmentation method is performed. The Cytokit parameters for segmentation with a membrane channel were specified as follows: memb_min_dist 1, memb_sigma 5, memb_gamma 0.25, marker_dilation 3, memb_propagation_regularization 0.25, memb_hole_size 20. Parameters for nucleus extension segmentation were memb_min_dist 8, memb_sigma 5, memb_gamma 0.25, marker_dilation 3. For dataset 9, we have memb_min_dist 4 and

marker_min_size 4. In addition to outputting the nuclei and cell segmentation masks, Cytokit provides information that describes the abundance of protein within each cell. More specifically, for each cell in the dataset and for each marker, Cytokit computes the average pixel intensity value over the nuclei segmentation mask and the cell segmentation mask. For the analysis done in this paper, we use the average value over the cell segmentation mask to compare different cell segmentation methods.

**Clustering and annotating cell types**. We performed dimensionality reduction and clustering using the Cellar[41] tool. We used UMAP as our embedding and Leiden clustering with a resolution of 0.3 and 15 nearest neighbors. We annotated the clusters using key markers for canonical cell types known for these tissues. Specifically, for datasets 8–10, we used pan-cytokeratin (PAN-CK) for epithelial cells, smooth muscle actin (SMActin), CD31, and LYVE-1 for lymphatic and vessel endothelial cells, and CD45 for immune cells of hematopoietic lineage. Among CD45+ cells, we further defined major cell subsets as follows: CD20+ B cells, CD3+CD4+ and CD3+CD8+ T cells, CD4+FOXP3+ regulatory T cells, and CD11c+CD68+CD15+ myeloid cells. Clusters denoted as proliferating cells expressed high Ki67. For dataset 11, major cell types were defined as follows: CD7+CD8+ T cells, PAX5+Ki67+CD34+ proliferating stem cells, CD31+ endothelial and stromal cells, CD45RO+CD25+CD3+Ki67+ memory activated T cells, CD68+CD163+ monocytes and macrophages, and CD3+FOXP3+CD25+CD4+ memory regulatory T cells. Some clusters were later merged based on these expert annotations. UMAP embeddings and channel montages colored by abundance of these key marker proteins are in Supplementary Figs. 11–15.

**Gating thresholds**. Thresholds for determining the presence/absence of proteins as summarized in Supplementary Table 5 were computed as follows. For each relevant protein, we calculate the mean ($\mu$) and standard deviation ($\sigma$) of the background pixel intensities. The background pixels are the pixels that are not part of any cell segmentation mask. The threshold is $\mu + 2\sigma$. Cells whose average intensity value over its segmentation mask (computed by Cytokit) is greater than the threshold are labeled as expressing that protein. The corresponding gating/biaxial plots can be found in Supplementary Fig. 8.

**Sequence-based methods for predicting membrane/surface markers**. For each labeled marker in the training set (Supplementary Table 3), we obtained scores from the SURFY[21], PrediSi[22], and SignalP 5.0[23] methods. For SURFY, we used the web database at http://wlab.ethz.ch/surfaceome and used the scores under the column 'SURFY score'. For PrediSi and SignalP 5.0, we used the web tools at http://www.predisi.de/ and http://www.cbs.dtu.dk/services/SignalP/, respectively, and input amino acid sequences for the proteins. Both of these methods output a score between 0 and 1 for the specified protein. These scores are compared to the true classification labels (Supplementary Table 3) to produce the curves in Fig. 3b.

**Spearman's rank correlation method**. The Spearman's rank correlation method from[19] selects membrane protein markers for a particular spatial proteomics dataset with the following steps: (1) compute the Spearman's rank correlation coefficient for each possible protein pair, (2) take the 10 protein pairs with highest correlation coefficient, (3) choose the 4 most frequent proteins present in those pairs as the final membrane markers. To compare this method with RAMCES, we assigned scores to proteins based on their frequency in the top 10 most correlated pairs, normalized between 0 and 1. The protein that appeared the most frequently in the top 10 received a score of 1, and any proteins not present in the top 10 received a score of 0. These scores are compared to the true classification labels (Supplementary Table 3) to produce the curves in Fig. 3b.

**Segmentation comparisons**. Supplementary Table 6 compares RAMCES segmentations that use combined top markers with segmentations that use only individual markers or the nucleus extension method. Supplementary Tables 7 and 8 compare RAMCES combined segmentations, individual marker segmentations, and nucleus extension segmentations with the two expert manual segmentations. Supplementary Table 9 compares the two expert manual segmentations with each other. Below are the definitions of the Jaccard index and Dice coefficient we use for these tables. $TP = $ # true positives, $FP = $ # false positives, $FN = $ # false negatives.

$$\text{Jaccard index} = \frac{TP}{TP + FP + FN} \tag{7}$$

$$\text{Dice coefficient} = \frac{2TP}{2TP + FP + FN} \tag{8}$$

**Reporting summary**. Further information on research design is available in the Nature Research Reporting Summary linked to this article.

## Data availability

The HuBMAP data used in this study are available in the HuBMAP data portal [https://portal.hubmapconsortium.org/] with HuBMAP IDs HBM869.VZJM.366 [https://portal.hubmapconsortium.org/browse/dataset/a6ccc344f88a164766d1251053173009],

HBM432.LLCF.677 [https://portal.hubmapconsortium.org/browse/dataset/75edcda4f3ff5bef72383d5d082438c2], HBM588.FHDS.363 [https://portal.hubmapconsortium.org/browse/dataset/8dd0ef5cafa3541cf9f0661db64662b7], HBM279.TQRS.775 [https://portal.hubmapconsortium.org/browse/dataset/077f7862f6306055899374c7807a30c3], HBM337.FSXL.564 [https://portal.hubmapconsortium.org/browse/dataset/f0c58e670ceb445e6ab02c6a20c83aee], HBM376.QCCJ.269 [https://portal.hubmapconsortium.org/browse/dataset/4514230f7473a496201a4e45c4ff9568], HBM754.WKLP.262 [https://portal.hubmapconsortium.org/browse/dataset/c95d9373d698faf60a66ffdc27499fe1], HBM556.KSFB.592 [https://portal.hubmapconsortium.org/browse/dataset/00d1a3623dac388773bc7780fcb42797], HBM288.XSQZ.633 [https://portal.hubmapconsortium.org/browse/dataset/f86b9efc87074bf03cd53932d8f1e76f]. Segmentation masks and RAMCES outputs are available in Zenodo with the identifier "https://doi.org/10.5281/zenodo.5655738"[42]. Celltype annotations can be found at the Cellar tool [https://data.test.hubmapconsortium.org/app/cellar][41]. Processed primary imaging data for the bone marrow dataset was from[7] from the 'Multi-tumor TMA' data, region 4. The primary imaging data for the mouse spleen dataset was from[1].

## Code availability

RAMCES is available at github.com/mdayao/ramces[43].

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

## Acknowledgements

Monica Dayao is a predoctoral trainee supported by the NIH T32 training grant T32 EB009403 as part of the HHMI-NIBIB Interfaces Initiative. Work was partially funded by NIH grant OT2OD026682 to Z.B.J. This work used the Bridges/Bridges-2 system, which is supported by NSF award number OAC-1928147 at the Pittsburgh Supercomputing Center (PSC). We thank Myles Bumgarner for the manual segmentation annotations provided. The results here are in whole or part based upon data generated by the NIH Human BioMolecular Atlas Program (HuBMAP).

## Author contributions

M.T.D. and Z.B.J. conceived the concepts behind RAMCES. M.T.D. developed the method, analyzed data, and wrote the paper. Z.B.J. supervised the project. M.B. and C.W. performed CODEX imaging experiments and provided biological annotations of results.

## Competing interests

The authors declare no competing interests.
