## [Peer Review File · Nature Communications]

REVIEWER COMMENTS

Reviewer #1 (Remarks to the Author):

“Membrane marker selection for segmenting spatial proteomics data” by Dayao et al describes a neural network driven approach (RAMCES) for unsupervised selection of markers with membrane localization. The method produces a ranked list of membrane markers and corresponding weights such that when combined an aggregated membrane channel could be produced to be used as an assisting (in combination with nuclear) or primary channel for the cell segmentation. Given the recent proliferation of “spatial biology” methods such as cyclic IF, cyclic IHC, CODEX, Mass Cytometry (MIBI and Hyperion), CellDive (Multiomix, GE), Zellkraftwerk and many others – this seems to be a very timely effort meant to deliver a tool enabling better and more universal segmentation pipelines so need by the spatial bio community. The authors base their efforts primarily on a number of unpublished, yet public CODEX data generated by HUBMAP consortium. They also generate their own CODEX data (scanning the murine thymus). This latter data is used for final validation of segmentation accuracy achieved by combining RAMCES and Cytokit.

Critique:

1. There is at the moment no universally accepted method to evaluate the quality of cell segmentation. The authors used detectable cell population frequencies to prove superior performance of segmentation enhanced by RAMCES. They claim that “The percentage of cells that co-express these sets of proteins increases when using RAMCES segmentation compared to the default nucleus extension segmentation from Cytokit, as expected for these tissues.” Yet no details on how this observation was done are provided. If it was done by gating – corresponding biaxial plots have to be shown. If it was done by clustering the biaxials confirming the coherence of identified cellular populations have to be provided. We don’t know to which extent the percentages reported by authors correspond to the bias of manual gating. More figures are needed. And needless to say – in many cases more isn’t better – it is unclear how having more cells within a gate or cluster proves that RAMCES boosts the segmentation quality

2. Figure 4 clearly shows an example of data with strong presence of imaging artifacts. The membrane markers are strongly shifted compared to nuclear (this is especially noticeable in the lower left panel of Fig.4). By default CODEX primary data is supposed to be properly drift compensated before it can be segmented. I examined the particular dataset that was used for figure 5 – the nuclei indeed seemed to be drift compensated between the cycles. Yet – strong misalignment (compare to nuclei) in some marker channels could be detected. This artifact could arise from unadjusted filter turret for a given channel (e.g. 530 or whichever was used for CD3e), from systematic z-shift due to non-adjusted optical pathway (in such cases channels will be acquired at a z slightly off compared to the nuclear z). All in all – perhaps due to yet unpublished status of

HUBMAP data or due to lack of training of the microscopist that made the dataset – the training set of the multiplexed imaging data (shown on Fig.4) wasn't of good quality. These kind of artifacts (perhaps not as aggravated as in the Fig4) are common for the ultra-multiplexed datasets – so one may argue that the authors deliberately chose the “problematic” cells to show that RAMCES enables to recover such cells otherwise improperly segmented. But then this has to be properly explained in the text. It would be nice if coordinate of cells shown by authors in Fig4 were provided – such that the cells could be found in the original HUBMAP data. This particular dataset (I went and checked the HUBMAP data) shows a very densely populated lymph node. The authors show a very non-typical part of it probably from the periphery – demonstrating just a couple of isolated cells. In summary Figure 4 deliberately or not is based on bad imaging data and does not sufficiently explain the point that authors want to make

3. I would like to see and examine the primary (drift compensated) data described in “Application to new CODEX datasets” section. Again – no gating data is shown. The authors provide multicolor Voronoi plot to prove that the segmentation quality is high. Yet – how do we know that the objects they identify indeed correspond to the cells the way they the identity is assigned. For example – the authors claim that a number of grey colored objects on their plots are Tregs. How do we know that? The authors need to show channel montages for a number of randomly picked “Tregs” such that we see that the cells their algorithm picks as Tregs are. CD45, CD4, Treg positive.

In conclusion. The effort represented in manuscript is meaningful and justified. Yet the proof provided is insufficient, incomplete and is based (possibly by intent, but then needs to be explained) on a bad case of CODEX imaging data. When corrected according to the comments above – this paper would be a great and valuable fit for a specialized cytometry journal – I don't think this paper especially as is – is a good fit for Nature Communications

Reviewer #2 (Remarks to the Author):

This manuscript describes a data processing pipeline for spatial proteomics that could be very helpful for experimentalists to analyse tissue data. The RAMSES system ranks protein markers to enable cell segmentation in CODEX data. Membrane proteins serve as markers to spatially localize the cell boundaries. The data flow chart in fig 1 identifies two classification stages - the ranking of the different proteins and the classification of membrane / non membrane proteins where the membrane proteins then provide information on cell boundaries.

Data processing pipelines should provide information how certain the final results are and how design changes in the selection of algorithms will impact the final outcome. The RAMSES segmentation pipeline suggests a novel classification scheme to identify suitable marker combinations and segments these data then using Cytokit. The essential ranking process serves the cell segmentation to support discovery of spatial patterns in tissue.

The manuscript has some deficits with respect to validation of individual steps in the pipeline; it should be improved to estimate error sources at the various steps of the processing pipeline, i.e., to provide uncertainty quantification to see the main influence factors for the final segmentation quality.

Major points:

1) The key methodological finding in this publication seems to be the ranking of the protein markers and their combinations that characterize the membranes of different cell types. Segmentation algorithms are then used as a scoring tool to quantify the usefulness of the marker combinations. It is unclear how the quality of the segmentation results are measured? Are the cell boundaries compared with pixel precision or are boundaries accounted for as in agreement when they both are within a small distance.

A clear description is missing if segmentation is considered to be an unsupervised task or if expert boundary annotations are available to judge the correctness of cell boundaries.

2) Various influence factors contribute to the final success of a segmentation pipeline in medical image processing. It would be helpful if the ultimate goal of this biomedical image analysis pipeline - the discovery of cell patterns in annotated images like in Fig 5 - could be related to the uncertainty of various algorithms in the processing pipeline. What is the influence of more robust ranking procedures on the segmentation quality? What is overfitting of the data analysis strategy compared to the final cell segmentation and the patterns that we like to discover?

3) The UMAP projection of the data into a low-dimensional visualization space (2-dim or 3-dim) defines a non-linear transformation. What guarantees can the authors give that the structures are properties of the data rather than artefacts of the visualization algorithm? This question should be addressed since the user might be misled by pattern that appear in the visualization but might not be supported by the data. It is known in multidimensional scaling (MDS) that dimensional mismatch,

i.e., visualization of high dimensional data in low-dimensional spaces, can generate ring like patterns when the method does not compensate for this dimension mismatch.

4) The paper would gain if the authors could add some uncertainty quantification for the different processing steps. This information is necessary to estimate false discovery rates of the cell identification

done by such a processing pipeline.

Minor points:

line 32: "optimla" => "optimal"

Fig. 3b: The precision recall curves are confusing or misleading if the usual definition for precision = $TP / (TP + FP)$ and recall = $TP / (TP + FN)$ [TP: true pos., FP: False pos., FN: false neg.] is used. A precision of 1 can only occur if $FP=0$; for close to zero recall this would imply that $TP \ll FN$, i.e. almost all TP have not been discovered. Therefore, decreasing precision recall curves are not a very encouraging sign for a data analysis method.

Reviewer #3 (Remarks to the Author):

Authors propose a new tool, RAMCES, to enhance cell segmentation from CODEX multiplexed imaging. Tool seems promising based on presented results, with an appropriate proposed methodology including CNN design, preprocessing and training steps, which are relatively well designed.

But there are some points where the manuscript deserves to be improved.

The way the manuscript is presented is misleading: the use of "proteomics data" in the title and manuscript is confusing, as the term "spatial proteomics". These words make thinking about the full proteome detected by mass spectrometry, while here this is more about immunohistochemistry and immunofluorescence microscopy for a few dozen of markers.

In introduction, authors propose to improve the existing tools by proposing a method generalizable to other tissues (line 31: "However, as we show, while it works well for some tissues it may not work well for others."). Here the paper only focuses on spleen, lymph and thymus. This study targets tissues related to the immune cells. The proposed method may work for other tissues, but it hasn't been tested. Maybe an immune-infiltrated tumor in an organ could have been welcomed.

Authors compare other existing tools performing cell segmentation. These tools present a very low AUC, which is surprising. Is it technically possible to use RAMCES on the data used by these other tools?

Number of samples is low, which may be a problem to properly assess the validity of models' performances. Authors have performed data augmentation, but is it the same data seen in different "point of view", not new data.

Minor:

- line 32: typo on "optimal"

- Line 221: Test dataset or Train dataset ?

- Line 49: expression is confusing here since there is not expression data

Reviewer 1 comments

“Membrane marker selection for segmenting spatial proteomics data” by Dayao et al describes a neural network driven approach (RAMCES) for unsupervised selection of markers with membrane localization. The method produces a ranked list of membrane markers and corresponding weights such that when combined an aggregated membrane channel could be produced to be used as an assisting (in combination with nuclear) or primary channel for the cell segmentation. Given the recent proliferation of “spatial biology” methods such as cyclic IF, cyclic IHC, CODEX, Mass Cytometry (MIBI and Hyperion), CellDive (Multiomix, GE), Zellkraftwerk and many others – this seems to be a very timely effort meant to deliver a tool enabling better and more universal segmentation pipelines so need by the spatial bio community. The authors base their efforts primarily on a number of unpublished, yet public CODEX data generated by HUBMAP consortium. They also generate their own CODEX data (scanning the murine thymus). This latter data is used for final validation of segmentation accuracy achieved by combining RAMCES and Cytokit.

Critique:

1. There is at the moment no universally accepted method to evaluate the quality of cell segmentation. The authors used detectable cell population frequencies to prove superior performance of segmentation enhanced by RAMCES. They claim that “The percentage of cells that co-express these sets of proteins increases when using RAMCES segmentation compared to the default nucleus extension segmentation from Cytokit, as expected for these tissues.” Yet no details on how this observation was done are provided. If it was done by gating – corresponding biaxial plots have to be shown. If it was done by clustering the biaxials confirming the coherence of identified cellular populations have to be provided. We don’t know to which extent the percentages reported by authors correspond to the bias of manual gating. More figures are needed. And needless to say – in many cases more isn’t better – it is unclear how having more cells within a gate or cluster proves that RAMCES boosts the segmentation quality.

We agree with the reviewer’s comment that in many cases more isn’t better, though for the tissues we analyzed (lymph node, spleen, thymus), it is expected that single positive T cells (CD3+CD4+, CD3+CD8+), double positive T cells (CD4+CD8+) and macrophages (CD68+CD4-CD8-) would be well represented¹². We thus focused on these cells in the comparisons we performed in the original submission.

Still, based on this comment and comments from Reviewer 3, we performed additional analysis to test the accuracy of the multi-channel segmentations. We first looked at the agreement between RAMCES segmentations using the weighted output and segmentation that only uses

¹ Parel, Yann, and Carlo Chizzolini. "CD4+ CD8+ double positive (DP) T cells in health and disease." *Autoimmunity reviews* 3.3 (2004): 215-220.

² Varol, Chen, Alexander Mildner, and Steffen Jung. "Macrophages: development and tissue specialization." *Annual review of immunology* 33 (2015): 643-675.

top individual channels (based on RAMCES dataset specific rankings). We show (Table S6) that RAMCES segmentations have, on average, much better overlap with all three top channels compared to their average pairwise overlap, highlighting the advantage of the weighted channel generated by RAMCES. We also performed expert manual segmentation on two tiles from a lymph node dataset and computed the overlap between the manual labeling and different segmentation methods using different marker sets. Manual segmentation was performed by referencing all of the image channels; however, we note that the DAPI and CD45 channels were most heavily relied on, which obviously impacted the results. Results, presented in Table S7 and below show that RAMCES segmentation improves over segmentation using two of the top three ranked channels and on the average of all three. Segmentation using only the 2nd ranked channel (which happens to be CD45) is in slightly better agreement with the manual annotations. This highlights the issue of ground truth for this case since any manual curated ground truth is likely to be biased by the person performing the segmentation. By combining the top three channels, RAMCES is able to overcome such biases and obtain results that are in good agreement with both expert curation and expected cell populations. As for the comment on gating, the cell population percentages presented in Table S5 were indeed calculated by gating. As suggested, we have now added the corresponding biaxial plots to the Supplementary (Fig S3).

The table below (also Table S7) shows cell segmentation mask overlap (calculated by the Jaccard index and Dice coefficient) with the expert manual annotation. Top2, top3 and top4 correspond to the RAMCES segmentation using the top 2,3,4 combined membrane markers. Highlighted in green is the segmentation used in the paper. Rank1, rank2 and rank3 correspond to the segmentation using the first, second and third-ranked **individual** (not combined) protein channels. The row highlighted in yellow is the average value of those individual protein segmentations. The row highlighted in red is the default nucleus extension segmentation method from Cytokit. It is shown that on average, the combined top 3 segmentation has greater with the manual segmentation than the individual channel or nucleus extension segmentations. Additionally, the performance between the combined top2,3,4 segmentations are similar, with Jaccard index and Dice coefficient values ± 0.002 within each other on average. However, we note that the segmentation using the individual second-ranked channel (CD45) performs best across the two tiles. This is likely because, as mentioned above, we referenced the CD45 channel the most often when manually segmenting the tiles.

Lymph node	Jaccard index			Dice coefficient		
	Tile 1	Tile 2	Avg	Tile 1	Tile 2	Avg
Top2 (combined) vs manual	0.6035	0.6985	0.6510	0.7527	0.8225	0.7876
Weighted (top3) vs manual	0.6043	0.6930	0.6487	0.7533	0.8187	0.7860

Top4 (combined) vs manual	0.6037	0.6972	0.6505	0.7529	0.8216	0.7873
Rank1 (CD4) vs manual	0.5917	0.6592	0.6255	0.7434	0.7946	0.7690
Rank2 (CD45) vs manual	0.6152	0.7095	0.6624	0.7618	0.8301	0.7951
Rank3 (CD20) vs manual	0.5607	0.6743	0.6175	0.7185	0.8054	0.7611
Avg rank# vs manual	0.5892	0.6810	0.6351	0.7412	0.8100	0.7756
Nucl-ext vs manual	0.5434	0.6569	0.6002	0.7042	0.7922	0.7482

2. Figure 4 clearly shows an example of data with strong presence of imaging artifacts. The membrane markers are strongly shifted compared to nuclear (this is especially noticeable in the lower left panel of Fig.4). By default CODEX primary data is supposed to be properly drift compensated before it can be segmented. I examined the particular dataset that was used for figure 5 – the nuclei indeed seemed to be drift compensated between the cycles. Yet – strong misalignment (compare to nuclei) in some marker channels could be detected. This artifact could arise from unadjusted filter turret for a given channel (e.g. 530 or whichever was used for CD3e), from systematic z-shift due to non-adjusted optical pathway (in such cases channels will be acquired at a z slightly off compared to the nuclear z). All in all – perhaps due to yet unpublished status of HUBMAP data or due to lack of training of the microscopist that made the dataset – the training set of the multiplexed imaging data (shown on Fig.4) wasn't of good quality. These kind of artifacts (perhaps not as aggravated as in the Fig4) are common for the ultra-multiplexed datasets – so one may argue that the authors deliberately chose the “problematic” cells to show that RAMCES enables to recover such cells otherwise improperly segmented. But then this has to be properly explained in the text. It would be nice if coordinate of cells shown by authors in Fig4 were provided – such that the cells could be found in the original HUBMAP data. This particular dataset (I went and checked the HUBMAP data) shows a very densely populated lymph node. The authors show a very non-typical part of it probably from the periphery – demonstrating just a couple of isolated cells. In summary Figure 4 deliberately or not is based on bad imaging data and does not sufficiently explain the point that authors want to make.

We thank the reviewer for this comment. Following this comment, we found that a subset of the datasets we analyzed were indeed not drift compensated. We note that this is not related to any data presented on the HuBMAP portal since that data was processed in a different manner than the processing discussed in this paper. The reason for the drift is the fact that for

some tiles we initially treated every channel separately to predict markers and so did not register them as Cytokit usually does. While this does not affect the selection of the top markers (since each one is evaluated using the DAPI image for its cycle independently), it did impact the combined channel since it can include a set of non-registered channels. We have now fixed this issue by performing registration prior to marker selection and channel integration. The table below shows the average absolute row and column translations (in pixels) performed on the top markers for each dataset. The average translations are within 1 pixel for most datasets. Note that this drift compensation issue *only* affected the RAMCES combined output and did not impact the methods we compared to, since these use a single channel approach. Following this correction, we re-calculated all the statistics in the paper for the different cell types and show that we still have an increase in the percentage of CD4+ T cells, CD8+ T cells, double positive T cells, and macrophages with the RAMCES segmentation compared to the default nucleus extension segmentation. We have updated figure (Fig 4), along with coordinates of the cells. Additionally, as requested we have added a figure showing a more densely packed example (Fig 4). We also performed analysis on a new dataset from a different tissue (bone marrow) and show that the method works for this tissue as well [Figure S5].

Average row and column drift compensation translation magnitude (absolute value, in pixels) across the top 3 ranked marker cycles for each evaluation dataset.

Dataset #	Avg row translation	Avg column translation
5 (lymph node)	0.640	0.587
6 (spleen)	0.550	0.693
7 (thymus)	1.028	1.017
8 (lymph node)	0.757	0.810
10 (thymus)	0.862	0.455
Overall average	0.767	0.712

3. I would like to see and examine the primary (drift compensated) data described in “Application to new CODEX datasets” section. Again – no gating data is shown. The authors provide multicolor Voronoi plot to prove that the segmentation quality is high. Yet – how do we know that the objects they identify indeed correspond to the cells the way they the identity is assigned. For example – the authors claim that a number of grey colored objects on their plots are Tregs. How do we know that? The authors need to show channel montages for a number of randomly picked “Tregs” such that we see that the cells their algorithm picks as Tregs are. CD45, CD4, Treg positive.

Gating was discussed above. As for channel montages, as suggested we have added these for some example cell types to verify the annotation of those cells (Figs 4, S10). We have also added figures of the UMAP embeddings colored by the abundance of chosen proteins to motivate the annotation of the cell types (Fig S6-S9).

In conclusion. The effort represented in manuscript is meaningful and justified. Yet the proof provided is insufficient, incomplete and is based (possibly by intent, but then needs to be explained) on a bad case of CODEX imaging data. When corrected according to the comments above – this paper would be a great and valuable fit for a specialized cytometry journal – I don't think this paper especially as is – is a good fit for Nature Communications.

As noted above, the revised manuscript improves on these issues in a number of ways including enhanced analysis of the accuracy of the resulting segmentation, improvement in the pre-processing of some of the datasets and analysis of data from additional tissues. All results indicate that RAMCES improves upon prior methods. Given the increase in use and importance of spatial, single cell proteomics data we believe that RAMCES will be a useful method for researchers using such data. Combined, we hope these improvements address the concerns raised by the reviewer.

Reviewer 2 comments

This manuscript describes a data processing pipeline for spatial proteomics that could be very helpful for experimentalists to analyse tissue data. The RAMSES system ranks protein markers to enable cell segmentation in CODEX data. Membrane proteins serve as markers to spatially localize the cell boundaries. The data flow chart in fig 1 identifies two classification stages - the ranking of the different proteins and the classification of membrane / non membrane proteins where the membrane proteins then provide information on cell boundaries.

Data processing pipelines should provide information how certain the final results are and how design changes in the selection of algorithms will impact the final outcome. The RAMSES segmentation pipeline suggests a novel classification scheme to identify suitable marker combinations and segments these data then using Cytokit. The essential ranking process serves the cell segmentation to support discovery of spatial patterns in tissue.

The manuscript has some deficits with respect to validation of individual steps in the pipeline; it should be improved to estimate error sources at the various steps of the processing pipeline, i.e., to provide uncertainty quantification to see the main influence factors for the final segmentation quality.

This is addressed in response to specific comment 2 below which elaborates on this point.

Major points:

1) The key methodological finding in this publication seems to be the ranking of the protein markers and their combinations that characterize the membranes of different cell types. Segmentation algorithms are then used as a scoring tool to quantify the usefulness of the marker combinations. It is unclear how the quality of the segmentation results are measured? Are the cell boundaries compared with pixel precision or are boundaries accounted for as in agreement when they both are within a small distance. A clear description is missing if segmentation is considered to be an unsupervised task or if expert boundary annotations are available to judge the correctness of cell boundaries.

We thank the reviewer for this comment. In the original submission, all analysis was indeed unsupervised and was based on known biology about the presence (or absence) of pairs or triplet of markers in cells in the tissues we analyzed. Following this comment, we extended the unsupervised analysis and have also added supervised analysis. For the former, we now show that RAMCES segmentations have the greatest overlap with each of the individual membrane channel segmentations when compared to the average agreement between these channels (Table S6). This is important in cases where different markers are more suitable for different cell types within the same image. The ability to use a combination of markers enables RAMCES to identify cells from different types than those identified with any single marker. For the supervised analysis, we performed expert manual segmentation of a portion of a lymph node dataset and computed the overlap between the manual labeling and the RAMCES

segmentation, the individual membrane channel segmentations, and the nucleus extension segmentation. Manual segmentation was performed by referencing all of the image channels; however, we note that the DAPI and CD45 channels were most heavily used when labeling. We show that on average, the RAMCES segmentation has the greatest overlap with the manual labeling (Table S7 and also on page 2 of this response). Note, of course, that the supervised segmentation is problematic since it is based on a human selection of a specific markers, which as noted above, can be a problem for some cell types or tissues. Still, the fact that the top marker and the average of all 3 top markers was lower than the combined RAMCES agreement provides support to the use of an automated pipeline for such analysis.

2) Various influence factors contribute to the final success of a segmentation pipeline in medical image processing. It would be helpful if the ultimate goal of this biomedical image analysis pipeline - the discovery of cell patterns in annotated images like in Fig 5 - could be related to the uncertainty of various algorithms in the processing pipeline. What is the influence of more robust ranking procedures on the segmentation quality? What is overfitting of the data analysis strategy compared to the final cell segmentation and the patterns that we like to discover?

We note that the specific segmentation toolkit/algorithm we used, Cytokit, is deterministic and so we cannot use it to evaluate different cutoffs for the segmentation masks themselves. Still, given this comment we evaluated the robustness and overfitting potential of RAMCES by repeating the analysis by combining the top 1, top 2, top 3 (which is what we use in the paper) and top 4 markers on the dataset on which we have performed expert manual segmentation. We show that the overlap between the manual segmentation and the RAMCES segmentation using the top 2, 3 and 4 markers is greater than the average overlap between the manual segmentation and individual top 3 channels or the nucleus extension segmentations (Table S7 and also on page 2).

We have also added channel montages and UMAP overlays of protein abundance to present the set of markers used to segment the different cells. These images show that no individual channel dominates the assignment which likely reduces overfitting (Figs S6-S9).

3) The UMAP projection of the data into a low-dimensional visualization space (2-dim or 3-dim) defines a non-linear transformation. What guarantees can the authors give that the structures are properties of the data rather than artefacts of the visualization algorithm? This question should be addressed since the user might be misled by pattern that appear in the visualization but might not be supported by the data. It is known in multidimensional scaling (MDS) that dimensional mismatch, i.e., visualization of high dimensional data in low-dimensional spaces, can generate ring like patterns when the method does not compensate for this dimension mismatch.

To help support the 2D embeddings generated based on the segmentations, we have now added figures to the supplement in which we present the abundance of some of the proteins which are known markers of specific cell types. As the figures show, for most clusters identified by the UMAP embedding we see enrichment of (different) cell type specific markers which

indicates that these clusters represent real biological groupings rather than artifacts related to the dimensionality reduction (Figs S6-S9).

4) The paper would gain if the authors could add some uncertainty quantification for the different processing steps. This information is necessary to estimate false discovery rates of the cell identification done by such a processing pipeline.

As suggested, we have now added results for different choices for the method. Specifically, we added results for using different number of markers (between 1 and 4). We show that results for using 2,3 and 4 markers are almost the same in terms of agreement with manual annotations (difference of 0.15%) whereas they all improve on using the average of the top 3 single markers (improvement of 2%) (Table S7).

RAMCES provides a score between 0 and 1 for each marker profiled which represents the average CNN model's output probability that the marker is a membrane marker. We can calculate the model's uncertainty of each marker using Shannon entropy. The scores and entropy values of the top 5 ranked markers have been added to Table S4.

We also note that the segmentation itself is based on Cytokit, which is a deterministic method, and so we cannot derive statistics on issues related to FDR for different cutoffs. We now explain this in Discussion.

Minor points:

line 32: "optimla" => "optimal"

Fixed.

Fig. 3b: The precision recall curves are confusing or misleading if the usual definition for precision = $TP / (TP + FP)$ and recall = $TP / (TP + FN)$ [TP: true pos., FP: False pos., FN: false neg.] is used. A precision of 1 can only occur if $FP=0$; for close to zero recall this would imply that $TP \ll FN$, i.e. almost all TP have not been discovered. Therefore, decreasing precision recall curves are not a very encouraging sign for a data analysis method.

We agree with the definition of precision and recall the reviewer provides. We are not sure why the reviewer claims that decreasing PR curves are not a positive result. In general, these curves usually begin at (0,1) or close to it (high precision, low recall) and end at (1,baseline), where 1 is the recall and baseline=fraction of positive examples in the dataset. Thus, a perfect classifier would have a curve that is at a right angle at point (1,1). Each point on the curve represents a different threshold (between 0 and 1) of the classifier and so decreasing curves are actually what we would like to see for a good classifier. We note that one of the curves in Figure 3b (the PrediSi curve) starts at (0,0) (low precision, low recall), which may have led to the comment above. However, this occurs because this classifier returns zero true positive examples for one of the tested thresholds.

Reviewer 3 comments

Authors propose a new tool, RAMCES, to enhance cell segmentation from CODEX multiplexed imaging. Tool seems promising based on presented results, with an appropriate proposed methodology including CNN design, preprocessing and training steps, which are relatively well designed.

But there are some points where the manuscript deserves to be improved.

The way the manuscript is presented is misleading: the use of "proteomics data" in the title and manuscript is confusing, as the term "spatial proteomics". These words make thinking about the full proteome detected by mass spectrometry, while here this is more about immunohistochemistry and immunofluorescence microscopy for a few dozen of markers.

We used a terminology that is commonly used in this domain, for example in³. To avoid confusion and following this comment we changed the title to 'Membrane marker selection for segmenting single cell proteomics data'.

In introduction, authors propose to improve the existing tools by proposing a method generalizable to other tissues (line 31: "However, as we show, while it works well for some tissues it may not work well for others."). Here the paper only focuses on spleen, lymph and thymus. This study targets tissues related to the immune cells. The proposed method may work for other tissues, but it hasn't been tested. Maybe an immune-infiltrated tumor in an organ could have been welcomed.

As suggested, we extended our analysis to test additional tissues. In addition to the tissues analyzed in the original paper, the revised version includes an analysis of bone marrow CODEX data using RAMCES. As we show, the method is able to successfully segment cells for that tissue as well. Please see Figs S5,S9.

Authors compare other existing tools performing cell segmentation. These tools present a very low AUC, which is surprising. Is it technically possible to use RAMCES on the data used by these other tools?

SURFY, PrediSi, and SignalP5 are methods trained on amino-acid sequencing data of proteins in order to classify signaling/membrane proteins. No imaging data was used to train or test them. Unlike these methods, RAMCES is trained on labeled imaging data; and of course, cannot be trained on sequence data only. Thus, it would not be possible to train or use RAMCES on the data that these other methods were trained on. This is exactly the point we tried to make, because these methods do not use the images and rely only on the sequencing data of the specified proteins, they lose the valuable context-specific information that the images provide. Therefore, it is not very surprising that these methods have a low AUC on the test dataset.

³ Lundberg, E., Borner, G.H.H. Spatial proteomics: a powerful discovery tool for cell biology. Nat Rev Mol Cell Biol 20, 285–302 (2019). <https://doi.org/10.1038/s41580-018-0094-y>

The only method for which imaging data is used for training is the Spearman correlation coefficient method that we are also comparing RAMCES to. This is an unsupervised method that uses the imaging data to choose a membrane protein. While this method was proposed in⁴ and used in the histoCAT cytometry toolbox⁵, it was never thoroughly investigated in those papers, and so direct comparison to the data used to evaluate it is only possible with the CODEX data we used for such comparison in the paper.

Number of samples is low, which may be a problem to properly assess the validity of models' performances. Authors have performed data augmentation, but is it the same data seen in different "point of view", not new data.

While the number of individuals from which samples are taken for training is indeed low (6), the number of images is quite large. Specifically, the RAMCES CNN model is trained on ~330,000 different 128x128 images. The main limitation is that RAMCES is trained and tested images from only human lymph node, spleen and thymus, and mouse spleen. We have attempted to address this by adding additional test data for other tissues (Fig S5). This brings the total number of test datasets to 7.

Minor:

- line 32: typo on "optimal"

Fixed.

- Line 221: Test dataset or Train dataset ?

Test dataset, but we have changed the sentence make it clearer.

- Line 49: expression is confusing here since there is not expression data

We removed 'expression' here.

⁴ Schüffler, Peter J., et al. Automatic single cell segmentation on highly multiplexed tissue images. *Cytometry Part A* 87.10 (2015): 936-942.

⁵ Schapiro, D., Jackson, H., Raghuraman, S. et al. histoCAT: analysis of cell phenotypes and interactions in multiplex image cytometry data. *Nat Methods* 14, 873–876 (2017). <https://doi.org/10.1038/nmeth.4391>

Reviewers' comments:

Reviewer #1 (Remarks to the Author):

I would like to acknowledge and commend authors of the "Ramces" manuscript for carefully reviewing and responding to suggestions outlined in my review of the initial version of their manuscript.

I have carefully read and examined the new version of the main text and of the supplementary figures. I believe that authors made a significant effort to answer my critique, but the following issues remain unresolved.

1. The authors implemented Jackard Index and Dice Coefficient scores to examine the overlaps of segmentation masks made by RAMCES based combined channel with masks produced either by single membrane channel driven segmentation runs or by manual curation. The RAMCES combined channel – based segmentations (Table S6) overlaps slightly (in the range of single digit percentages) better with the rest of segmentation masks than different single channel results with each other. Importantly – this result does not really prove that RAMCES produces BETTER segmentation results. As a proof of that a detailed look at Fig.4 bottom panel shows that barely a single cell is outlined precisely by all of the methods used. Rather what we see is that all the segmentation methods suffer from more or less similar biases precluding them from producing a precise outline, and somehow – RAMCES combined channel averages the errors contributed by the individual segmentation channels. The overlap between masks produced by different computational methods cannot therefore be used as a criterion to judge about the quality of segmentation. Not to mention that at a "ball park" estimated segmentation precision of 70-80% , 2%-5% increase in segmentation quality is considerably below the noise level. In resolution of issues raised above – perhaps one possible interesting unbiased method of segmentation performance evaluation would be to examine correlation of image reconstructed from the segmentation mask with an original image – this way the "golden" unbiased reference is the primary image itself.

2. In comparison with manually curated outlines – all methods performed equally poor. The average TableS7 value is 0.7. Implying (exactly as mentioned by me above) the 70% precision and hence 30% noise – which is well above the benefit in overlap contributed by RAMCES.

4. The biaxial plots and the gates applied to these plots (Figure S3) suggest that the gates are placed "manually" – such as for example in Figure S3c lower left CD8/CD4 plot. As there is not clear separation between the peaks – It is not clear which criterion was used to place the gate.

5. No access to primary drift compensated image data with segmentation masks and cell annotations was provided (requested by me in the initial review of the manuscript).

6. Finally – there is an inherent danger in aggregation of membrane channels for the means of segmentation. Many membrane proteins bear significant presence inside Golgi and endoplasmic reticulum and rather mark the cytoplasm than purely the membrane. For that matter careful selection of individual membrane channel devoid of this property (vesicular presence) may be more beneficial than combination of several membrane channels which may result in aggregation of artifacts (both biological and technical) associated with these channels.

In summary. I believe that this manuscript as is deserves publication in a specialized journal where skillful and targeted expert community will have a chance for critical decision making (as to whether this method should or should not be applied).

I don't think this manuscript is a good fit for a broad audience journal such as Nature Communications.

Reviewer #2 (Remarks to the Author):

Comments on resubmission NCOMMS-21-12973A

The authors have responded to my comments in a constructive way and I am mostly satisfied with the revision. I particularly appreciate the extra effort to validate the results of this biomedical image processing pipeline.

I assume that the precision_recall curves in fig 3b refer to segmentations of cells in tissue images. The TP are the number of true positive cell boundaries and FN denotes the number of boundary pixels that are not part of a segment boundary. With a recall of 0.6, we achieve only a precision of 0.6. That means that you have $TP = 3/2 FP$ and these false positives give rise to erroneous segment boundaries. It would be good if the authors could relate these precision-recall curves to images of segmentation boundary detections like in fig 4.

Minor points for further consideration:

I 354: The revision describes "expert annotations". I couldn't find the information how many experts have independently annotated the images? Experts often differ in their annotations and their uncertainty contains information on the reliability of the gold standard.

Since the authors don't have "ground truth" in for image segmentation (I 131), the authors should define what they considered as the "gold standard" in this study. What could be done is to define a consensus segmentation of annotations from 2-3 annotators to avoid overfitting to only one expert.

I 216: The fact that an algorithmic procedure is deterministic, does not mean that the uncertainty cannot be estimated. For example the authors could use two/three images of the same tissue sample or, if not available, a second sample can be generated by resampling a measured image and imputing the non-sampled pixels. Such a technique is related to bootstrap but applied to images. The posterior estimate of the segmentation is based on the randomness in the input and not due to random bits in the algorithmic execution.

I 193: "that by the using ..." -> "that by using ..."

I 199: "... manually-curated ground truth is likely biased by the individual performing the segmentation."

How can this statement be justified if we are lacking any comparison memasure for ground truth?

I am convinced that we have to list good arguments when we claim that results are biased due to an annotation process. Without a reference for comparison, such a claim appears to me as vacuous.

In fig 3, it would be informative for the reader to see which ROC curve points and precision recall values have been calculate and which are linear interpolations of the plotting program.

Reviewer #3 (Remarks to the Author):

Authors have addressed all my comments.

I have no other suggestions

Reviewer #1 (Remarks to the Author):

I would like to acknowledge and commend authors of the “Ramces” manuscript for carefully reviewing and responding to suggestions outlined in my review of the initial version of their manuscript.

I have carefully read and examined the new version of the main text and of the supplementary figures. I believe that authors made a significant effort to answer my critique, but the following issues remain unresolved.

1. The authors implemented Jackard Index and Dice Coefficient scores to examine the overlaps of segmentation masks made by RAMCES based combined channel with masks produced either by single membrane channel driven segmentation runs or by manual curation. The RAMCES combined channel – based segmentations (Table S6) overlaps slightly (in the range of single digit percentages) better with the rest of segmentation masks than different single channel results with each other. Importantly – this result does not really prove that RAMCES produces BETTER segmentation results. As a proof of that a detailed look at Fig.4 bottom panel shows that barely a single cell is outlined precisely by all of the methods used. Rather what we see is that all the segmentation methods suffer from more or less similar biases precluding them from producing a precise outline, and somehow – RAMCES combined channel averages the errors contributed by the individual segmentation channels. The overlap between masks produced by different computational methods cannot therefore be used as a criterion to judge about the quality of segmentation. Not to mention that at a “ball park” estimated segmentation precision of 70-80% , 2%-5% increase in segmentation quality is considerably below the noise level. In resolution of issues raised above – perhaps one possible interesting unbiased method of segmentation performance evaluation would be to examine correlation of image reconstructed from the segmentation mask with an original image – this way the “golden” unbiased reference is the primary image itself.

We note two points regarding this comment. First, we have not attempted to optimize the segmentation accuracy since this was not the major focus of the paper. Specifically, we have used the same segmentation method used by HuBMAP (Cytokit) for all samples. And for this, we have shown that using RAMCES results improves the segmentation. Second, these comments and the entire discussion of segmentation accuracy that underlie them are somewhat sidetracking the main point of the paper. RAMCES is not a new segmentation method but rather, as we emphasize in the title and abstract, a method to select markers for segmentation analysis. This is a new problem that only arises for CODEX and other single cell spatial proteomics platforms. The advantage of these new platforms, which is what RAMCES exploits, is that multiple optimal markers can now be selected *after* performing the experiments and not beforehand.

Still, given the above comment we performed additional analysis to test whether RAMCES segmentation results are indeed better than the other methods we compared to and the human annotation we obtained. While no ground truth information is available, we compared

the disagreement portions of different segmentation methods. Specifically, we looked at pixels in the image which RAMCES claims are cells (RAMCES = 1) and other methods claim that they are not (0) and at pixels which RAMCES assigns to background whereas the other segmentation methods assign them to cells. We would expect that successful methods would have an average biomarker distribution for areas where the method assigns to inside cells and the comparison method assigns to background similar to the distribution for areas where both methods assign to inside cells. See Figure S4 in the revised manuscript for a visual representation of the areas.

Fig 1 and Fig 2 below summarize the agreements and disagreements between the RAMCES segmentations (using the RAMCES combined output with the top 3 ranked markers) and the default nucleus extension segmentations for dataset 5 as labeled in Supplementary Table S1. In both plots, the y-axis is showing the average pixel intensity of a biomarker, labeled on the x-axis. In the first plot (Fig 1), the blue bars show the average pixel intensity in the areas where the two segmentation methods agree in the foreground (fg, inside of the cells). The orange bars show the segmentation agreement between the two segmentation methods in the background (bg, outside of the cells). The green bars show the segmentation disagreement. In the second plot (Fig 2), we show the average pixel intensity of biomarkers where RAMCES segmentations label a cell and the nucleus extension segmentations do not (purple bars, 'disagree ramces=1') and where the nucleus extension segmentations label a cell and RAMCES segmentations do not (pink bars, 'disagree nuclext=1'). The blue and orange bars are the same across Fig 1 and Fig 2, showing the agreement in the foreground and the total disagreement, respectively.

Fig 1 highlights that there is increased biomarker signal in pixels identified as inside cells by both of the segmentation methods when compared to those determined to be outside cells, as we would expect. Fig 2 shows that in areas which RAMCES labels as inside of cells and the nucleus extension method does not, biomarker signal resembles that of levels seen for cell (foreground) areas. In contrast, areas where the nucleus extension method labels as cells and RAMCES does not look much more similar to background areas. This result suggests that the RAMCES segmentations capture more of the biomarker signal than the nucleus extension method, which we can interpret as a measure of improved segmentation.

Plots in Fig 3 summarize the disagreements between RAMCES and our manual expert segmentations in a similar way to the comparison in Fig. 2. As in Fig 2, the blue bars show the average biomarker intensities in areas where the respective segmentation methods agree in the foreground, and the orange bars show average intensities where they agree in the background. Yellow bars show the average intensities in disagreement areas where the manual segmentation labels inside of cells. The purple bars in Fig 3 show the average intensities in areas where the RAMCES segmentation labels inside of cells and the manual segmentation does not. Fig 3 shows that for all biomarkers except DAPI, the RAMCES segmentations actually agrees better with cell biomarker signal than the manual segmentations.

We have added Figs 1-3 to the supplement (Figures S5-S7).

Biomarker intensities for agreement and disagreement between nuclxt and ramces segmentations

Biomarker intensities disagreements between nuclxt and ramces segmentations

Biomarker intensities disagreements between manual and ramces segmentations

2. In comparison with manually curated outlines – all methods performed equally poor. The average TableS7 value is 0.7. Implying (exactly as mentioned by me above) the 70% precision and hence 30% noise – which is well above the benefit in overlap contributed by RAMCES.

We disagree with the reviewer comment that the agreement we observed for RAMCES and the expert annotations we obtained is not good. A recent paper (Hou et al, Scientific Reports 2020, PMID: 32561748), which performed comprehensive analysis of human segmentation annotation (for a different type of data), states that ‘Based on our patch-level quantitative assessment, compared to manual segmentation ... the nucleus segmentation data has an average Dice coefficient of least 77% ... These results are similar to the inter-annotator agreement in our experiments’. Thus, Dice coefficients of 75-79% agreement are good given the disagreement between human annotators themselves. Additionally, a paper that investigated the accuracy of crowdsourcing segmentation annotations (Gurari et al., AAAI Conference on Human Computation & Crowdsourcing 2016) showed that the median Jaccard score of biomedical image annotations was 0.85 compared to their defined gold standard.

To further determine if the agreement we observed is good, we added another expert annotator in order to determine the agreement between two manual annotations and compare it to the agreement we presented between the expert and RAMCES. As we show in Table 2 below and in Table S9 in the revised manuscript, the average Dice coefficient between the two expert annotators is 0.6987, which is slightly better than the agreement with RAMCES and in line with prior work as discussed above. The RAMCES segmentations still improve our results on average compared to the default nucleus extension method and segmentations produced by using individual markers as the membrane markers (Table 1 below, same format as Table S7 in the manuscript). Table 1 has been added to the supplement (Table S8).

Table 1: Cell segmentation mask overlap (calculated by the Jaccard index and Dice coefficient) with the second expert manual annotation. Top2, top3 and top4 correspond to the RAMCES segmentation using the top 2,3,4 combined membrane markers, respectively. Rank1, rank2 and rank3 correspond to the segmentation using the first, second and third-ranked individual (not combined) protein channels.

	Jaccard Index			Dice Coefficient		
	Tile 1	Tile 2	Avg	Tile 1	Tile2	Avg
Top2 (combined)	0.5110	0.4752	0.4931	0.6764	0.6442	0.6603
Top3 (combined)	0.5110	0.4926	0.5018	0.6764	0.6600	0.6682
Top4 (combined)	0.5099	0.4837	0.4968	0.6752	0.6520	0.6636
Rank1	0.5019	0.4711	0.4865	0.6683	0.6405	0.6544
Rank2	0.5184	0.4926	0.5055	0.6828	0.6600	0.6714

Rank3	0.4932	0.4661	0.4797	0.6606	0.6358	0.6482
Avg rank#	0.5045	0.4766	0.4906	0.6706	0.6454	0.6580
Nucl-ext	0.4347	0.3956	0.4152	0.6060	0.5669	0.5865

Table 2: Cell segmentation mask overlap between the two experts' annotations.

	Jaccard Index			Dice Coefficient		
	Tile 1	Tile 2	Avg	Tile 1	Tile 2	Avg
Agreement between the manual annotations	0.5570	0.5174	0.5372	0.7155	0.6819	0.6987

4. The biaxial plots and the gates applied to these plots (Figure S3) suggest that the gates are placed “manually” – such as for example in Figure S3c lower left CD8/CD4 plot. As there is not clear separation between the peaks – It is not clear which criterion was used to place the gate.

This was explicitly mentioned in the Supplement and Methods section. We state in the caption of Figure S3 (S8 in the revised version) and in the Methods section (‘Gating thresholds’) how we calculate the thresholds; they are based on the background pixel distribution of the images.

5. No access to primary drift compensated image data with segmentation masks and cell annotations was provided (requested by me in the initial review of the manuscript).

The primary drift-compensated image data without our segmentation masks and annotations are available on the HuBMAP portal (portal.hubmapconsortium.org/) using the HuBMAP IDs found in Table S1. Datasets 8-10 have now been added to the portal, and we have updated Table S1 with their IDs accordingly.

As for the segmentation masks, we have uploaded these to <https://doi.org/10.5281/zenodo.5655737>. Celltype annotations can be found on the Cellar tool: <https://data.test.hubmapconsortium.org/app/cellar>. We would like to note that Reviewer 1 did not explicitly request for these items in their initial review, only for the “the primary (drift compensated) data described in ‘Application to new CODEX datasets’ section”.

6. Finally – there is an inherent danger in aggregation of membrane channels for the means of segmentation. Many membrane proteins bear significant presence inside Golgi and endoplasmic reticulum and rather mark the cytoplasm than purely the membrane. For that matter careful selection of individual membrane channel devoid of this property (vesicular presence) may be more beneficial than combination of several membrane channels which may result in aggregation of artifacts (both biological and technical) associated with these channels.

This is a new point that was not raised in the initial review, but it is still addressed in the manuscript. RAMCES is supervised and so can easily overcome the problem the reviewer mentions. The main goal of RAMCES is to *select appropriate markers*. Thus, markers that, as the reviewer notes, mainly target non-membrane areas can be easily avoided by training RAMCES on positive (purely membrane) and negative (Golgi and endoplasmic reticulum) proteins. This is exactly why we need a supervised method to select the markers rather than sequence or other automated selection method. Alternatively, if the user knows a-priori that specific biomarkers should not be used as a membrane marker during segmentation, they can be removed from the RAMCES ranking and not combined in the final RAMCES output by setting the `--exclude` option when running the code. We have emphasized these points in the Discussion of the revised manuscript.

In summary, I believe that this manuscript as is deserves publication in a specialized journal where skillful and targeted expert community will have a chance for critical decision making (as to whether this method should or should not be applied).

I don't think this manuscript is a good fit for a broad audience journal such as Nature Communications.

Reviewer #2 (Remarks to the Author):

Comments on resubmission NCOMMS-21-12973A

The authors have responded to my comments in a constructive way and I am mostly satisfied with the revision. I particularly appreciate the extra effort to validate the results of this biomedical image processing pipeline.

We thank the reviewer for this comment and his overall recommendation to accept the paper.

I assume that the precision_recall curves in fig 3b refer to segmentations of cells in tissue images. The TP are the number of true positive cell boundaries and FN denotes the number of boundary pixels that are not part of a segment boundary. With a recall of 0.6, we achieve only a precision of 0.6. That means that you have $TP = 3/2 FP$ and these false positives give rise to erroneous segment boundaries. It would be good if the authors could relate these precision-recall curves to images of segmentation boundary detections like in fig 4.

We believe that we may have confused the reviewer regarding the results presented in Figure 3. These are not for segmentation accuracy but instead for (binary) classification of the membrane protein markers (1 for a membrane marker and 0 for non-membrane proteins). We have now emphasized this in the 'RAMCES identifies combinations of membrane markers' section in Results and in the Methods ('Sequence-based methods for predicting membrane/surface markers' and 'Spearman's rank correlation method') and have also revised the caption of Figure 3 to emphasize this difference.

Minor points for further consideration:

I 354: The revision describes "expert annotations". I couldn't find the information how many experts have independently annotated the images? Experts often differ in their annotations and their uncertainty contains information on the reliability of the gold standard. Since the authors don't have "ground truth" in for image segmentation (I 131), the authors should define what they considered as the "gold standard" in this study. What could be done is to define a consensus segmentation of annotations from 2-3 annotators to avoid overfitting to only one expert.

We had 1 expert annotate the images; however, in light of this comment we evaluated another expert's annotations for the same images. We observe similar results in terms of agreement with RAMCES for the 2nd annotator (Table S8). In addition, as we show in Table 2 above and in Table S9 in the revised manuscript, the average Dice coefficient between the two annotators is 0.6987, in line with prior work that looked at agreement between human annotators for cell segmentation paper (Hou et al, Scientific Reports 2020, PMID: 32561748). The RAMCES segmentations still improve our results on average for the 2nd annotator compared to the default nucleus extension method and segmentations produced by using individual markers as the membrane markers (Table 1 above, same format as Table S7 in the manuscript).

I 216: The fact that an algorithmic procedure is deterministic, does not mean that the uncertainty cannot be estimated. For example the authors could use two/three images of the same tissue sample or, if not available, a second sample can be generated by resampling a measured image and imputing the non-sampled pixels. Such a technique is related to bootstrap but applied to images. The posterior estimate of the segmentation is based on the randomness in the input and not due to random bits in the algorithmic execution.

While we agree that we can perform bootstrap analysis for the images, since there is no ground truth for the segmentation it would be hard to determine uncertainty in results for the overall method. Still, to address this comment we performed bootstrap analysis on the membrane marker classification model (first part of RAMCES) by training 100 models on different bootstrap training samples. ROC and PR curves for the corresponding testing sets are shown in the figure below. The light gray lines show the curves for each individual bootstrap, and the green lines are the mean curves. The model achieves a mean AUC-ROC of 0.848 and AUC-PRC of 0.714, which is slightly better than the cross-validation results we reported in the manuscript itself (Figure 3a). The figure below has been added to the Supplementary (Figure S1).

I 193: "that by the using ..." -> "that by using ..."

Fixed

I 199: "... manually-curated ground truth is likely biased by the individual performing the segmentation."

How can this statement be justified if we are lacking any comparison memasure for ground truth?

I am convinced that we have to list good arguments when we claim that results are biased due to an annotation process. Without a reference for comparison, such a claim appears to me as vacuous.

There are actually several papers that explicitly discuss this issue and very convincingly show that expert annotation is biased. For example, a recent paper (Hou et al, Scientific Reports 2020, PMID: 32561748), which performed comprehensive analysis of human segmentation annotation (for a different type of data), states that 'Based on our patch-level quantitative assessment, compared to manual segmentation ... the nucleus segmentation data has an average Dice coefficient of least 77% ... These results are similar to the inter-annotator agreement in our experiments'. This clearly indicates that different experts would have different results for the same dataset. We have added this reference to the claim listed by the reviewer.

In fig 3, it would be informative for the reader to see which ROC curve points and precision recall values have been calculate and which are linear interpolations of the plotting program.

We have provided the files used to create these curves at this link: <https://doi.org/10.5281/zenodo.5655737>. The figure is already pretty busy, so we decided not to include it in the manuscript itself.

Reviewer #3 (Remarks to the Author):

Authors have addressed all my comments.
I have no other suggestions